

# Extracting small deformation beyond individual station precision from dense GNSS networks in France and Western Europe

Christine Masson[1], Stephane Mazzotti[1], Philippe Vernant[1], Erik Doerflinger[1]

1 Géosciences Montpellier, CNRS, University of Montpellier, Université des Antilles, Montpellier, 34000, France

*Correspondence to*: Christine Masson (christine.masson@umontpellier.fr)

**Abstract.** We use dense geodetic networks and large GPS datasets to extract regionally coherent velocities and deformation rates in France and neighboring Western Europe. This analysis is combined with statistical tests on synthetic data to quantify the deformation detection thresholds and significance levels. By combining two distinct methods, Gaussian smoothing and k-means clustering, we extract horizontal deformations with a 95% confidence level *ca.* 0.1–0.2 mm yr[-1] (*ca.* 0.5–1 x 10[-9] yr[-1])

on spatial scales of 100–200 km or more. From these analyses, we show that the regionally average velocity and strain rate fields are statistically significant in most of our study area. The first order deformation signal in France and neighboring Western Europe is a belt of N-S to NE-SW shortening *ca.* 0.2–0.4 mm yr[-1] ($1-2 \times 10^{-9}$ yr[-1]) in central and eastern France. In addition to this large-scale signal, patterns of orogen-normal extension are observed in the Alps and the Pyrenees, but methodological biases, mainly related to GPS solution combinations, limit the spatial resolution and preclude associations

with specific geological structures. The patterns of deformation in western France show either tantalizing correlation (Brittany) or anti-correlation (Aquitaine Basin) with the seismicity. Overall, more detailed analyses are required to address the possible origin of these signals and the potential role of aseismic deformation.

## 1 Introduction

The Global Navigation Satellite System (GNSS) is a primary dataset to study present-day crustal deformation, for example

through the computation of strain rate tensors, both in active tectonics areas (e.g., Indonesia or Greece; Gunawan et al., 2019; Chousianitis et al., 2015) and in very low deformation areas (e.g., Eastern Canada or India; Tarayoun et al., 2018; Banerjee et al., 2008). However, the analysis of regional and local deformation is commonly restricted by several factors, such as the precision of individual GNSS velocities, the presence of non-tectonic transient signals or the methods used to compute strain rates on different spatial scales (e.g., Cardozo et al., 2009; Zhu et al., 2011; Carafa and Bird, 2016). In

particular, the precision of individual GNSS velocities is a strong limitation in intraplate regions, where the amplitude of the tectonic signal is of the same order of magnitude as the measurement uncertainties (e.g., Calais and Stein, 2009; Tarayoun et al., 2018).

In this study, we evaluate the ability to extract regionally coherent and statistically significant information on the present-day deformation rates in France and neighboring Western Europe from GNSS networks. This analysis is performed in several



stages. (1) We first compute a consistent velocity dataset based on GPS (Global Positioning System) signals for over 900 stations, using statistical semi-automatic technics for time series processing (in particular offset and outlier detections). (2) The raw GPS velocities are analyzed using two independent methods (k-means clustering and Gaussian smoothing) in order to extract coherent velocities and deformations on spatial scales of 100–200 km. (3) We perform Monte Carlo and Bootstrap

sampling analyses on the original GPS velocities and on synthetic velocity data to estimate the detection threshold and the significance level of the computed velocity and strain rate fields.

Our study is focused on France and neighboring Western Europe since they represent an ideal location for testing these methodological developments. Indeed, they are often considered as a domain without significant deformation except in its bordering mountain ranges, the Alps (e.g., Houlié et al., 2018; Brockmann et al., 2012) and the Pyrenees (e.g., Neres et al.,

2018; Rigo et al., 2015). The Alps and Pyrenees have a high level of seismic activity (Fig. 1), allowing us to study how it relates to the GPS deformation rates, both in spatial distribution and style. The rest of France experiences a low to moderate diffuse seismicity in many regions (Cara et al., 2015; Fig. 1) suggesting that it undergoes a small but non-zero present-day deformation. We show that, in most of our study area, horizontal deformation rates can be estimated from GPS data with a 95% confidence level *ca.* 0.1–0.2 mm yr$^{-1}$ (*ca.* 0.5–1 x 10$^{-9}$ yr$^{-1}$) on a spatial scale of 100–200 km or more. We discuss the

relationship between the observed deformation and regional seismicity and neotectonic indicators.

## 2 GPS networks and data analysis

We process data of 987 GNSS stations over a period ranging from 1998 to the end of 2016. The time series cover time spans from 3 to 18 years with an average duration of 7 years. All stations have not been installed under the same conditions and the same goals (geodetic, cadastral, …). About a quarter of the stations are installed and maintained by private networks, for

which we do not have all the information on the site monumentation or history of equipment changes. Figure 2 shows the distribution of these stations and identifies the additional campaign stations and Swiss stations combined to the main dataset (Section 2.4).

For data processing, we use the Precise Point Positioning (PPP) software developed by the Canadian Geodetic Survey of Natural Resources Canada (CSRS-PPP v1.05) (Héroux and Kouba, 2001). Optimum processing parameters and options to

25 compute 24 h-average daily positions are defined by Nguyen et al (2016), following International GNSS Service (IGS) products and recommendations (Kouba et al., 2009; Dow et al., 2009). We use the IGS Final Products for satellite orbits and clocks, satellite and ground antenna absolute phase center mapping, and Earth rotation parameters. We also use standard models for tropospheric delay corrections (VMF1, Boehm et al 2006) and solid Earth and ocean tide loading corrections (FES 2004, Lyard et al., 2006).



### 2.1 Time series analysis

Daily position time series are modeled with a constant velocity, annual and semi-annual periodic motions, instantaneous offsets, and random colored noise:

$$x(t) = vt + A_1 \sin(\omega_1 t + \phi_1) + A_2 \sin(\omega_2 t + \phi_2) + C_i . H(t, T_i) + \varepsilon \quad (1)$$

where $x$ is the daily position, $t$ is the time, $v$ is the velocity, $A_{1,2}$, $\omega_{1,2}$ and $\varphi_{1,2}$ are the amplitudes, periods and phases of the annual and semi-annual motions, $C_i$ and $T_i$ are the amplitude and date of the $i^{th}$ offset (with $H$ the Heaviside function) and $\varepsilon$ is the residual. To jointly estimate these parameters (except for the noise parameter), we use a linear least-square

inversion of the model (Eq. 1).

Because only a small percentage of stations is associated with reliable equipment logs that could provide position offset dates, the dates of potential offsets are automatically detected according to the method described in Masson et al. (2019). Their statistical analysis shows that this method compares favorably with most automatic and manual detection methods (Gazeaux et al., 2013), with an average detection level of about 52% vs. about 20% of false positives. Overall, using this

method results in horizontal (resp. vertical) velocity biases are smaller than 0.2 mm yr$^{-1}$ (resp. 0.5 mm yr$^{-1}$) at 95% confidence levels for series longer than 8 years (Masson et al., 2019).

We calculate the velocity standard errors using Williams et al. (2003) generic expression for colored noise with a non-integer spectral index. The spectral index and amplitude of the colored noise are estimated using a least-square inversion of the residual ($\varepsilon$, eq. 1) spectrum limited to periods between 1/12 and T/2 years (with T the length of the time series). This simple

approach only provides a first-order estimation of the noise parameters and velocity standard errors (compared to a more complex non-linear method, such as maximum likelihood). In agreement with other recent studies (Santamaria-Gomez et al., 2011; Nguyen et al., 2016), we observe that the spectral indices vary from -0.8 to -0.4, indicating a combination of white and flicker noise. Using synthetic time series analyses, Masson et al. (2019) show that the simple "least-square spectrum" approach yields estimations of velocity standard errors that are reasonable for series with a spectral index smaller than -0.6

but that are underestimated for series with a spectral index greater than -0.4. The latter corresponds to only 18% of our data, allowing us to have confidence in the calculated velocity standard errors.

### 2.2 Common-mode spatial filtering

In order to identify and correct for non-tectonic transient signals and noise common to the whole network, we define a common-mode correction (Wdowinski et al., 1997) by stacking the residual time series of 31 stations with near-complete

data and durations greater than 17 years. Such a correction typically allows adjusting for network-wide biases such as systematic orbit and clock issues due to the IGS combination process, or potential large-scale environmental loads. On average, the application of the common-mode correction to all stations reduces the daily dispersion on the North, East and





Up components from 1.94 mm to 1.36 mm (-30%), 2.49 mm to 2.07 mm (-17%) and 4.54 mm to 4.12mm (-9%) respectively. The stacked time series show a decrease in dispersion of the North and East components between 1998 and 2002 (cf. Fig. A), which is likely related to improvements in IGS satellite ephemerids and reference frame definition (Griffiths, 2019). The latter can be observed on the North, East and Up components directly at or shortly after the transitions
to IGS08 and IGb08 reference frames (cf. Fig. A).

Pluri-annual signals with amplitudes of 1–5 mm over periods of 2–5 yr are evidenced in the North, East and Up components (e.g., 2008–2011). Similarly, residual annual and semi-annual signals are also detected despite the integration of these periodic motions in the time series model (eq. 1). These residual signals indicate the presence of coherent phenomena operating over large geographical scales (100s km) with varying durations, regularities and cyclicities. Longer series are
required to confirm the presence of decadal or pluri-decadal (periods 10–15 years) signals, which could impact the estimations of long-term velocities.

### 2.3 Reference Frame

After PPP processing and the application of the regional common-mode correction, daily positions and velocities are in an informal reference frame tied to the IGS satellite orbits and clocks (IGb08). In order to analyze the relative motions and
deformations in France and conterminous Western Europe, we rotate the velocities in a local frame, first using the Eurasia-ITRF2014 rotation defined by the ITRF2014 (Atamimi et al., 2016), then by minimizing the residual velocities of 200 stations distributed evenly over our region of interest and with time series duration greater than 7 years (Fig. 2).

In this "France-centered" reference frame, two regional-scale systematic motions are detected in the velocity field (Fig. 3a). In northwestern Spain, a northward motion of *ca.* 0.1–0.3 mm yr$^{-1}$ may be associated with the clockwise rotation of Iberia
relative to Eurasia described by Palano et al (2015) and Neres et al., (2016). This motion is not observed in the northeastern part of Spain, likely reflecting the location of the rotation pole and the specific dynamics of the Pyrenees (see Section 5.2). In northern Italy, the northern motion increasing eastward from 0 to ca. 3 mm yr$^{-1}$ is compatible with the counterclockwise rotation of the Adria microplate relative to Eurasia, with a rotation pole located in the Po plain (Battaglia et al., 2004; Farolfi et al., 2015). These tectonic movements are beyond the scope of our study.

### 2.4 Network densification

In order to densify the velocity field, we integrate data from additional networks (including campaign data) that were either processed or analyzed with a different procedure from the main dataset.

### 2.4.1 The Swiss network

Raw (RINEX) data are not available for Swiss stations, leading us to directly combine the processed velocities
(http://pnac.swisstopo.admin.ch/restxt/) with ours. We perform a six-parameter Helmert transformation (rotation and translation rates without scale) to minimize the residual velocities at the 58 sites common to the two datasets. Post-





transformation statistics indicate an average agreement for the common sites of 0.15, 0.13, and 0.37 mm yr$^{-1}$ in the North, East and Up components. The Swiss velocities include both campaign and permanent station velocities, for which the uncertainties are estimated with an unspecified method (Brockmann et al., 2012). These uncertainties are much lower than ours, requiring a first-order scaling to homogenize the two datasets. By comparing the distributions of uncertainties of the 58

common sites between the two solutions, we estimate a multiplication factor 24 for the North and East components and 20 for the Up component, to adjust the distribution of Swiss uncertainties to ours. This simple scaling yields a good first-order agreement of the two distributions (based on the 10, 50, and 90 quantiles), but detailed analysis still points out residuals issues that affect the interpretation of the combined field (see Section 5.3).

### 2.4.2 Campaign data

For the French Alps, we include campaign data from sites measured for at least 48h during surveys in 1998, 2004, 2015, 2017 and 2018. The characteristics of the shortest and longest time series are 14 years with 2 surveys and 21 years with 5 surveys. In 1998 and 2004, measurements were made on tripods, while since 2015 the sites have been modified to use anchored mast in order to optimize installation stability and antenna height measurement. We use the same data processing for 24-h daily positions as for permanent stations but the time series analysis is different. Since the data are sporadic (a few

points every 4–10 years), it is impossible to model annual and semi-annual seasonal signals, detect offsets and estimate noise characteristics by spectral analysis. A careful re-analysis of the antenna heights and RINEX data allows us to estimate a long-term velocity (including daily position weighting by the measurement duration) for 29 sites in the French Alps (Fig. 2). In the Pyrenees, campaign data extend from 1992 to 2017, but because of the relatively low quality of satellite orbits and clocks in the early and mid 1990s (Griffiths, 2019), data before 1996 cannot be included in the PPP processing, which

represents ca. 20% of the campaign sites. To avoid this problem, we process the 75 Pyrenees campaign sites with the GAMIT/GLOBK software (Herring et al. 2009) following the strategy defined by Rigo et al. (2015) to process Pyrenean GPS data from 1992 to 2010. Given the early measurement dates, few continuous GPS sites are available to provide the reference frame stabilization. We identify 17 permanent sites with data back to 1995, in common with our PPP solution allowing a combination of the two velocity fields. These sites are unevenly distributed, with three-quarters located northeast

of the Pyrenees within a few 100s km, leading to potential reference biases that are discussed in the velocity field interpretation (see Section 5.2).
Because of their sporadic and sparse point density, campaign time series are not amenable to noise model estimation and computation of formal uncertainties. There is no standard method for estimating campaign velocity uncertainty, but most of them rely scaling based on the nearby continuous GNSS sites (Tarayoun et al., 2018; Beavan et al., 2016; Reilinger et al.,

2006). In this study, we use synthetic time series constructed from permanent-station parameters (Masson et al., 2019) to generate synthetic campaign time series with the Alps and Pyrenees campaign characteristics (dates and number of surveys). We generate datasets with 2–7 campaigns of 3 days for a variety of time spans. The deviations of the estimated velocities (compared to the true velocity set to 0 mm yr$^{-1}$) are used to estimate statistical standard errors on the campaign data. For time





series longer than 15 years with more than 2 surveys, the average horizontal velocity standard error is *ca.* 0.3 mm yr⁻¹. Longer durations and additional surveys reduce this standard error to *ca.* 0.2 mm yr⁻¹. We assign to each campaign site a velocity standard error accordingly to its campaign characteristics. This method yields identical uncertainties for sites of the same campaigns, thus neglecting inter-site variability, but it provides the most consistent campaign uncertainties compared

with the permanent data, allowing for the most efficient integration of the two datasets.

## 2.5 Statistical detection of outliers

GNSS sites can be affected by non-linear signals related to site conditions (e.g., monument instability, changes in site conditions) or transient local phenomena (e.g., resource extraction). It is necessary to identify and exclude these stations from velocity solutions aiming at a tectonic and geodynamic study. To do so, we use two statistical methods based on

geographical coherence considering that stations with a velocity significantly different from its neighbors is not representative of a tectonic movement (at least on the 100-km scale that we consider here). The statistical outlier detections are applied only on the horizontal velocities because vertical velocities show too much variability for robust results.

First, a regression tree analysis (Breiman et al., 1984) is used to divide the network into an un-prescribed set of rectangular regions presenting the minimum dispersion of the station variables (longitude, latitude, and North and East velocities

weighted by their standard errors). For all regions, we calculate the velocity range and then we determine the median of all ranges. Regions with a range larger than the overall median are subject to an outlier rejection. For each of those, sites with velocities outside of the ( $Q_1 - 1.5 \times IQR$ - $Q_3 + 1.5 \times IQR$ ) range are rejected, where *IQR* is the interquartile range, and $Q_1$ and $Q_3$ are the first and third quartiles (Tukey, 1977).

Second, outlier stations are detected according to the Mahalanobis distance criteria. Mahalanobis distances ($D_M$) are

computed for the whole network using each station variables (longitude, latitude, and North and East velocities) to define the barycenter, covariance, and 95% confidence interval of the multidimensional space formed by the network variables (Mahalanobis, 1936). Stations for which $D_M$ is greater than the network 95% confidence interval are considered as outliers and rejected.

Of the initial 1163 permanent stations, 180 stations are rejected as outliers. This outlier population has velocity statistics ($Q_1$,

median and $Q_3$) three times larger than the remaining station population. The final velocity field is shown in Figure 3. The station spatial density over France and immediately neighboring regions is on average 8.0 x 10⁻⁴ site/km²,(i.e., *ca.* 25 sites within a radius of 100 km), with differences between the regions (Fig. 2). Some regions such as the Paris Basin have a high density of stations (10.5 x 10⁻⁴ site/km²) while others have a lower density of stations, such as the Aquitaine Basin (6.3 x 10⁻⁴ site/km²). The integration of campaign stations has different impacts on the density in the French Alps and Pyrenees (Fig. 2).

In the former, the density of stations is above the network average with and without the campaign data (12.6 x 10⁻⁴ *vs.* 11.2 x 10⁻⁴ site/km²). On the contrary in the Pyrenees, the density is significantly below the national average without the campaign





stations ($5.0 \times 10^{-4}$ site/km$^2$). The resolution of the spatial coherence analysis on velocities may be lower in areas with low density of stations than in areas with high density of stations.

## 3 Extraction of coherent regional velocities

In our local "France-centered" reference frame, individual station velocities and uncertainties tend to be of the same order of
magnitude (Fig. 3), precluding a detailed interpretation of the kinematics and deformation. We use two independent statistical approaches (clustering and spatial smoothing) to extract spatially coherent information on the present-day deformation at a spatial scale of 100–200 km. To a first order, this distance represents the expected scale of tectonic deformation in France and neighboring regions. It corresponds to the average width of the major tectonic systems (French Alps, Pyrenees, South Armorican Shear Zone, etc.) as well as the spatial scale of interseismic deformation on a fault with a
seismogenic thickness of 10–25 km (Savage and Burford, 1973).

The smoothing and clustering methods are tuned to extract coherent signals on geographical domains at this spatial scale by minimizing local noise without losing the regional signal. The robustness of the results is estimated with two independent approaches:

- We use synthetic data to estimate each method detection level. The synthetic dataset corresponds to random velocities derived from the combination of a null long-term velocity, annual and semi-annual sinusoids, position offsets and colored noise (eq. 1), with all parameters based on our GPS data characteristics (cf. Masson et al., 2019 for details). Thus, this dataset corresponds to null velocity and strain rate fields with additional noise representative of our actual GPS data. Synthetic and actual data are compared after processing by the clustering or smoothing methods to estimate their respective detection levels.
- We estimate statistical standard errors on the velocities and strain rates calculated with the clustering or smoothing methods using a simple Monte Carlo and Bootstrap resampling of the original GPS velocities on the basis of their standard errors (Monte Carlo) and regional distribution (Bootstrap). Hereafter, the smoothed or clustered velocities and strain rates are given as the mean and 95% confidence interval ($CI_{95}$) of the Monte Carlo / Bootstrap samples. Using 95% confidence intervals (rather than the more common standard errors) yields a strict definition of
significant vs. not significant signals, which should be nuanced in cases of values close to the limits.

### 3.1 Clustering

Clustering is a parametric unsupervised learning method and has been used in GNSS analyses to perform geographical groupings of stations with optimum velocity consistency (Savage et al., 2013 and 2018; Liu et al., 2015; Ozdemir et al., 2019). Several methods of clustering exist; here we use the k-means method (Hartigan and Wong, 1979) whose advantage is
that the formed groups do not have a predefined geometry or size and allow for abrupt changes so to adapt to local geometries. In order to extract spatial coherence in the velocity field, we define clusters using station coordinates (longitude and latitude) and horizontal (north and east) or vertical velocities; vertical velocities are treated independently because of their large dispersions and uncertainties. The variables are not normalized, in order to ensure first-order geographical clustering with secondary adjustments based on the velocity variability.





The analysis takes place in four steps: 1) choice of random clusters (based on a predefined initial number); 2) minimization of the Euclidian distances between the cluster centroids and the different observations; 3) shift of the initial centroids to the mean of the groups; 4) minimization of the Euclidian distances according to new centroids from an increase of the cluster number and modifications of their boundaries. The last three steps are repeated until convergence to the predefined final

number of clusters and no observation changes clusters. At the end of the process, each GNSS station is attributed a velocity that corresponds to its final cluster median.

We run the algorithm 50000 times in order to take into account three stochastic aspects of the computation:

- The choice of the final number of clusters is crucial as it controls the average cluster size (geographical extent and number of stations). In order to average velocities over 100–200 km spatial scale, we vary the final number of
clusters between 75 and 200 (uniform distribution), resulting in a mean cluster area of 27.5 x $10^3$ km$^2$ (i.e., scale *ca.* 166 km), with variations of a factor of 2–3 in the cluster sizes. A larger number of clusters (smaller spatial scale) leads to strong variability and incoherence of the final velocities.
- The random choice of initial clusters results in variability in the final results. We take advantage of this effect to improve the statistics of the results by varying the initial number of clusters between 1 and the final number divided
by 2 (uniform distribution).
- In order to account for each station specific velocity uncertainty, we redefine its velocity using a random draw in a normal distribution defined by its velocity mean and standard deviation. Thus, sites with high uncertainties have a large variability and less weight in the overall cluster analysis.

For each station, the final velocity is defined as the median, with the associated 95% confidence interval, of the 50000

computations. The clustered horizontal and vertical velocities are presented on Figures 4 and 5a. The comparison of clustered velocities based on the actual GPS data and on the synthetic dataset indicates a detection level for the horizontal velocities of 0.12 mm yr$^{-1}$ (cf. Fig. Ba). Only 10% of the stations are associated with a clustered velocity below this detection level (Fig. 4a). Vertical velocities are not analyzed with the synthetic comparison because of their high dispersion and spatial noise. Their interpretation is thus subject to caution.

Coherent horizontal patterns are observed with velocities *ca.* 0.1–0.4 mm yr$^{-1}$ (Fig. 4a). Associated 95% confidence intervals are mostly *ca.* 0.1–0.2 mm yr$^{-1}$, except in the Pyrenees, Alps and Paris Basin where they reach 0.4–0.5 mm yr$^{-1}$ (Fig. 4b). The vertical velocities are associated with larger $CI_{95}$ *ca.* 0.4–0.8 mm yr$^{-1}$. The only significant vertical patterns (Fig. 5a) are the subsidence in the Paris Basin (*ca.* 0.5–1 mm yr$^{-1}$) and uplift in the Western and Central Alps (*ca.* 0.5–2 mm yr$^{-1}$).

**3.2 Gaussian smoothing**

Smoothing is a standard technique for data interpolation, filtering and noise reduction. It allows the extraction of a continuous field, minimizing small-scale noise and precluding abrupt spatial changes. We use a Gaussian smoothing function to compute at any point in space a smoothed velocity $V_i$ based on all GPS velocities weighted according to their standard errors and their distance to the computation point (Mazzotti et al., 2011):



$$V_i = \left( \sum_{n=1}^{N} \frac{G_n}{\sigma_{ni}^2} V_{ni} \right) / W_i \qquad (2)$$

*with*

$$G_n = e^{-\log(2) \frac{\Delta_n^2}{r_g^2}}$$

$$W_i = \sum_{n=1}^{N} \frac{G_n}{\sigma_{ni}^2}$$

where $i$ is the velocity component (North, East or Up), $r_g$ is the smoothing length (half-width of the Gaussian function), $V_n$ and $\sigma_n$ are the velocity and standard error of the GPS station $n$, and $\Delta_n$ is the distance to the GPS station $n$. The spatial derivative of the Gaussian function can be used to compute the smoothed horizontal strain rate tensor $\dot{\varepsilon}_{ij}$ at the same location:

$$\dot{\varepsilon}_{ij} = \frac{\log(2)}{r_g^2} \left[ \begin{array}{l} \left[ \left( \sum_{n=1}^{N} -\Delta_j \frac{G_n}{\sigma_{ni}^2} V_{ni} \right) / W_i - \left( \sum_{n=1}^{N} -\Delta_j \frac{G_n}{\sigma_{ni}^2} \right) \frac{V_{nj}}{W_j} \right] \\ + \left[ \left( \sum_{n=1}^{N} -\Delta_i \frac{G_n}{\sigma_{nj}^2} V_{nj} \right) / W_j - \left( \sum_{n=1}^{N} -\Delta_i \frac{G_n}{\sigma_{nj}^2} \right) \frac{V_{ni}}{W_i} \right] \end{array} \right] \qquad (3)$$

where $\Delta_{(i,j)}$ is the distance to the GPS station $n$ in the east or north direction.

15 The smoothing length $r_g$ controls the correlation length of the computed velocity and strain rate fields. To first order, this correlation length can be associated with the Gaussian smoothing scale $D_g = 2 \times r_g$, which corresponds to the distance within which the GNSS stations contribute 67% of the computed velocity (95% contribution within $2D_g$).

In order to extract deformation signals at the 100–200 km scale with minimum contribution from local noise, we estimate an optimal smoothing scale on the basis of a series of tests on the synthetic velocity data. Smoothing scales $D_g \leq$ 120–140 km

20 result in smoothed velocities that can reach up to 0.1–0.2 mm yr$^{-1}$ (compared to the true null velocity), whereas scales $D_g >$





140 km are associated with smoothed velocities smaller than 0.1 mm yr$^{-1}$ (cf. Fig. C). Hereafter, we present results based on the smoothing scale $D_g$ = 160 km, keeping in mind that results with $140 > D_g > 200$ km are similar at better than 0.1 mm yr$^{-1}$. The smoothed velocity and horizontal strain rate fields, and associated 95% confidence intervals, are presented in Figures 5b, 6 and 7. Horizontal strain rates are expressed in terms of "maximum strain rate":

$$e_{max} = max(|e_1|, |e_2|, |e_1 + e_2|) \qquad (4)$$

where $e_1$ and $e_2$ are the principal components of the strain rate tensor. The comparison of the smoothed velocities and strain rates based on the actual GPS data with those based on the synthetic dataset (cf. Fig. Bab) indicates detection levels of 0.04

10     mm yr$^{-1}$ for the horizontal velocities and 0.35 x 10$^{-9}$ yr$^{-1}$ for the horizontal strain rates. Less than 5% of the results are below the detection level (Figs. 6a and 7a), similar to the clustering analysis.

Coherent patterns are observed with horizontal (resp. vertical) velocities *ca.* 0.1–0.4 mm yr$^{-1}$ (0.2–1.6 mm yr$^{-1}$), similar to a first order to those obtained with the clustering method (Figs. 4 vs. 6 and Fig. 5). Statistical 95% confidence intervals on the horizontal velocities and strain rates are respectively *ca.* 0.05–0.1 mm yr$^{-1}$ and *ca.* 0.7–0.9 x 10$^{-9}$ yr$^{-1}$, indicating that more

than two-thirds of the velocities and strain rates are statistically significant (Figs. 6 and 7).

### 3.3 Coherence of clustering and Gaussian smoothing results

To first order, the clustered and smoothed velocity fields show a striking agreement in both the horizontal components (Figs. 4a and 6a) and the vertical components (Fig. 5) components. In order to better characterize the coherence of the two velocity fields, and its potential regional variability, we compute the difference between the two methods for each velocity

component (North, East and Up). The dispersion of the North and East differences (cf. Fig. D) illustrates the consistency of the two methods: 28% of the differences are smaller than 0.05 mm yr$^{-1}$ and 60% are smaller than 0.10 mm yr$^{-1}$. In Figure 8, we identify the areas where the horizontal velocity differences are larger than 0.1 mm yr$^{-1}$: the westernmost part of Brittany, parts the Paris Basin, most the Western Alps (especially near the France-Swiss-Italy borders) and the Pyrenees and their northern foreland. For these areas, the interpretations of the velocity fields must be carried out with caution. These velocity

differences at the level of 0.1–0.3 mm yr$^{-1}$ can be related to either abrupt spatial variations in the actual velocity and deformation patterns, or biases and limitations in one or both methods.

### 4 Large-scale deformation patterns

The analysis of the deformation in France and conterminous regions can be guided using the uncertainty levels described in section 3. To a first order, we consider as "well resolved" areas of 100–200-km scale or more within which (1) the GNSS

station density is similar to or larger than the network average, (2) the clustered and smoothed velocities are consistent at 0.1 mm yr$^{-1}$ or better. In contrast, areas where one of the criteria is not satisfied are considered less well constrained. These





correspond roughly to the region northeast of the France-Belgium-Germany border (very low station density), the center and northwest of the Paris Basin, most of the Alps and northwestern Italy, and the Spain-Pyrenees region south of about 44° N (Fig. 8). The last three are discussed in more details in section 5.

Most of central, northern and western France is associated with coherent clustered and smoothed velocities resolved at *ca.*
0.1 mm yr$^{-1}$. There, the first order and most important deformation signal is that, relative to the "France-centered" reference, western and central France are characterized by eastward to northeastward motions of 0.1–0.2 mm yr$^{-1}$, whereas northeastern France shows 0–0.2 mm yr$^{-1}$ southward to southwestward velocities (Figs. 4 and 6). The resulting overall deformation pattern corresponds to a belt of N-S to NE-SW shortening of $(1–2) \times 10^{-9}$ yr$^{-1}$ in central and eastern France (Fig. 7).

In details, this first-order kinematic pattern is associated with regional variability, with extension in the Armorican Massif
and a complex strain rate pattern in the Aquitaine Basin. In more details, from West to East:

- The western Armorican Massif is associated with a small E-W extension rate (0.4–0.8 × 10$^{-9}$ yr$^{-1}$, $CI_{95}$ = 1.0–1.2 × 10$^{-9}$ yr$^{-1}$) subject to caution because of the high uncertainty. The smoothed and clustered velocities differ, likely due to the network configuration (lack of stations on three sides of the peninsula), precluding a more detailed spatial analysis. In the eastern part, the extension rotates to a N-S direction with similar rates and slightly lower
uncertainties (0.3–0.7 × 10$^{-9}$ yr$^{-1}$, $CI_{95}$ = 0.6–0.8 × 10$^{-9}$ yr$^{-1}$). Smoothed and clustered vertical velocities reveal a low generalized subsidence (-0.2 mm yr$^{-1}$, $CI_{95} \approx$ 0.1 mm yr$^{-1}$ and -0.3 mm yr$^{-1}$, $CI_{95} \approx$ 0.3 mm yr$^{-1}$, respectively). The overall extension is roughly compatible with the deformation and stress analyses from focal mechanisms that indicate NE-SW extension in Brittany (Mazabraud et al., 2004).
- To the south, the Aquitaine Basin shows a complex pattern of high strain rates (1.5–2 × 10$^{-9}$ yr$^{-1}$) with N-S
shortening in the northwest, E-W shortening in the southwest in the Pyrenees foreland, and E-W extension in the east at the border with the Massif Central (Fig. 7). All rates are statistically significant, but the low density of GNSS stations in the east strongly limits the validity of the E-W extension pattern. Both smoothed and clustered vertical velocities show a barely significant small subsidence (-0.15 mm yr$^{-1}$, $CI_{95} \approx$ 0.1 mm yr$^{-1}$ and -0.3mm yr$^{-1}$, $CI_{95} \approx$ 0.4 mm yr$^{-1}$, respectively). The strong and spatially varying strain rates are surprising in light of the low seismicity (Fig.
1) and very few indications of active tectonics (Baize et al., 2013ab; Jomard et al., 2017ab). Peculiar hydrological loading is not reflected in the time series annual or pluri-annual signals.
- To the east, the large-scale pattern of E-W to NE-SW shortening is observed in the Massif Central (0.8–1.0 × 10$^{-9}$ yr$^{-1}$, $CI_{95}$ = 0.5–0.7 × 10$^{-9}$ yr$^{-1}$), associated with near-zero vertical velocities, and in northeastern France, including strong shortening rates (2–3 × 10$^{-9}$ yr$^{-1}$) rotating from E-W in the Bresse to N-S in the Upper Rhine Graben (Fig.
7). There, clustered velocities show a small subsidence (-0.4 mm yr$^{-1}$, $CI_{95} \approx$ 0.6 mm yr$^{-1}$) not present in the smoothed velocities, indicating either a small-scale signal (< 10s km) or noise in the original data. This deformation pattern is discussed in more detail in section 5.3.

Finally, Corsica presents a coherent, near-rigid northward motion relative to the continent, similar between the smoothing and clustering methods at 0.4 mm yr$^{-1}$ ($CI_{95} \approx$ 0.1 mm yr$^{-1}$), that may be related to the present-day tectonics of the Apennine-
Tyrrhenian system (Figs. 4 and 6). This northern motion creates a domain of significant NNW-SSE shortening of 2.0 × 10$^{-9}$ yr$^{-1}$ ($CI_{95} \approx$ 0.2 × 10$^{-9}$ yr$^{-1}$) north of Corsica (Fig. 7), compatible with the tectonic and seismicity observations along the Ligure Margin (Larroque et al., 2012, 2016).





**5 Regions with complex velocity and deformation patterns**

Complex velocity and deformation patterns require more careful analysis in three specific regions: the Paris Basin, the Pyrenees and the Western Alps (associated with the Upper Rhine Graben). The first is associated with a very low seismicity, unlike the other two that are the most seismically active area of Western Europe.

**5.1 The Paris Basin**

For 20% of the stations in the Paris Basin, the differences between the clustered and smoothed velocities are between 0.1 and 0.3 mm yr$^{-1}$ (Fig. 8), but there is a general consistency in the spatial variations of the velocity directions: from west to east, ENE to SE for smoothing, Fig. 6a, and NE to SSE for clustering, Fig. 4a. The spatial variations are by definition sharper for the clustering results. These result in a significant NE-SW shortening ($0.8 \times 10^{-9}$ yr$^{-1}$, $CI_{95} \approx 0.6 \times 10^{-9}$ yr$^{-1}$), part of the
large-scale shortening belt (section 4), without seismicity and with very few identified active faults (Fig. 1). Smoothed and clustered vertical velocities show a generalized subsidence (-0.3 mm yr$^{-1}$, $CI_{95} \approx 0.1$ mm yr$^{-1}$ and -0.5 mm yr$^{-1}$, $CI_{95} \approx 0.7$ mm yr$^{-1}$, respectively) with however a potential small area of uplift around Paris for the clustered velocities (0.35 mm yr$^{-1}$, $CI_{95} \approx 0.6$ mm yr$^{-1}$). The amplitudes of seasonal signals are more important than the national median (+29% for the annual signals in E and +48% in U) suggesting that hydrological processes or natural resource extraction may contribute to the
observed deformation.

**5.2. Pyrenees**

Most of the GNSS sites in the Pyrenees are campaign sites with large uncertainties. For about 50% of the stations, the differences between the smoothed and clustered velocities are between 0.1 and 0.4 mm yr$^{-1}$, but the orientations are similar (Fig. 4a and 6a). These differences may be due to the fact that most of the sites have been surveyed only twice with the first
survey in the mid 1990s, when the satellite orbits and clocks were of lower quality, and the permanent GPS network was very sparse providing only few common stations for the solution combination (see Section 2.4.2). As a result, the campaign velocities are associated with large standard errors, leading to a low weight in the smoothing results.

Despite these differences, a first-order pattern of N-S extension is clearly visible in both the clustered and smoothed velocities (between 0.2 and 0.4 mm yr$^{-1}$). The extension rate decreases from west ($1.5 \times 10^{-9}$ yr$^{-1}$, $CI_{95} \approx 0.8 \times 10^{-9}$ yr$^{-1}$) to
east ($1.0 \times 10^{-9}$ yr$^{-1}$, $CI_{95} \approx 0.8 \times 10^{-9}$ yr$^{-1}$), with an associated rotation of the extension direction from NNE-SSW to N-S (Fig. 9). Most vertical velocities are not significant, except in the eastern region with a value barely significant of 0.18 mm yr$^{-1}$, $CI_{95} \approx 0.2$ mm yr$^{-1}$ for the smoothed velocities.

The seismicity of the Pyrenees is heterogeneously distributed along the orogen (Chevrot et al., 2011; Calvet et al., 2013). It is diffuse in the east and more focused in the west along the North Pyrenean Fault system (Fig. 9). In the central part of
Pyrenees, the GPS-based extension zone extends over a wider area than the seismicity, but remains located within the





northern and southern limits of the Pyrenean orogen. This difference in the spatial distribution of the seismicity, orogen topography, and GPS extension may be attributed to three potential causes:

1. Actual present-day deformation only takes place within the zone of high seismicity and our GPS analysis does not allow such a fine localization because of the limits in the smoothing and cluster spatial scales (100–200 km, see Section 3).
2. The location of the current seismicity does not reflect the location of the deformation on a longer time scale, which is in contrast, captured by GPS data. This would indicate potential large earthquakes outside present-day seismicity zone.
3. The deformation measured by GPS at the surface reflects a deeper aseismic deformation centered on the orogen center / highest topography, and not only the crustal seismic deformation.

### 5.3. The Alps and the Upper Rhine Graben

In the Alps, for 28% of the stations the differences between the clustering and smoothing methods are between 0.1 and 0.5 mm yr$^{-1}$, but the velocities have consistent directions. Roughly northward velocities of 0.2–0.4 mm yr$^{-1}$ in the Swiss Alps and nearby Jura (Fig. 4a and 6a) result in significant N-S extension *ca.* $2.1 \times 10^{-9}$ yr$^{-1}$ near and south of the Swiss-Italy border, and N-S shortening *ca.* $1.8 \times 10^{-9}$ yr$^{-1}$ in the Vosges - Upper Rhine Graben - Black Forest region. The region in between (most of Switzerland) is associated with low deformation (Fig. 7). The western foreland of the Alps (Rhone Valley) is mainly characterized by E-W shortening (*ca.* $0.8–2 \times 10^{-9}$ yr$^{-1}$), in relation with the eastward motion of the Massif Central and central France (see Section 4). The central and southern parts of the Western Alps show a transition from N-S extension to strike-slip and E-W extension in the south, associated with the 0.2–0.4 mm yr$^{-1}$ eastward motion in southeastern-most France and the western Po Plain (Figs. 4a and 6a). To a first order, these deformation patterns in the French Alps are consistent with those observed in Walpersdorf et al. (2018), although our strain rate amplitudes are on average 2–3 time smaller, potentially due to the lower number of stations and spatial coverage used in Walpersdorf et al. (2018).

The vertical velocities (Fig 5) have the same large-scale pattern in the two methods, with a maximum of uplift in the northern French and Swiss Alps (*ca.* 1.5–2 mm yr$^{-1}$) decreasing to near-zero (± 0.5 mm yr$^{-1}$) in the Southern Alps, the Rhone valley, the Po Plain and the Jura. This pattern is consistent with that derived from previous GPS data analysis and combinations with leveling data (Nocquet et al., 2016; Sternai et al., 2018). A more detailed analysis of the transition from relatively fast uplift to near-zero (or small subsidence) rates is restricted by the limits of the smoothing and clustering methods, as well as the low precision of individual stations.

The seismicity is distributed over the whole Alpine system (Figs. 1 and 10), with a concentration of epicenters in the southwest of Switzerland (Deichmann et al., 2012). The GPS-derived N-S extension near the Swiss-Italy-French borders is compatible with the earthquake focal mechanism analysis of Delacou et al., 2004, but the peak of deformation is located south of the seismicity concentration in a region of relatively low activity (Fig. 10). The location of the maximal extension depends on how the uncertainties from the Swiss velocity field are scaled when combined with our solution. If the weight of the Swiss velocities is reduced, the maximum extension is shifted 50 km northward, near the Switzerland-Italy border at a location consistent with the seismicity. This issue points out the importance of consistent processing and uncertainty analysis





of all raw GNSS data. In addition, the region of Aosta is devoid of stations in our analysis, leading to an additional source of uncertainty on the precise localization of the maximum deformation. If the GPS-seismicity localization difference proves to be real, it may the expression of deformation not only accommodated by seismicity or of a lack of large earthquakes in the high strain rate area.

## 5 6. Conclusions

The application of semi-automatic statistical methods allows us to evaluate the potential of dense networks and large GNSS datasets to extract spatially coherent and significant deformation rates in France and conterminous Western Europe. In particular, the combination of spatial smoothing and clustering approaches on more than 900 GPS velocities results, for the first time, in the definition of horizontal velocity and strain rate fields with a 95% confidence level *ca.* 0.1–0.2 mm yr$^{-1}$ (*ca.*

0.5–1 x 10$^{-9}$ yr$^{-1}$) on spatial scales of 100–200 km or more (Figs. 4b, 6b, 7b). In most of the study area, the calculated velocity and strain rate patterns are just at or above the 95% significant level. Based on this analysis, several conclusions can be drawn:

- The first order and most important deformation signal in France and neighboring Western Europe is a belt of N-S to NE-SW shortening *ca.* 0.2–0.4 mm yr$^{-1}$ (1–2 × 10$^{-9}$ yr$^{-1}$) in central and eastern France.
- We observe orogen-normal (radial) extension *ca.* 1–2 × 10$^{-9}$ yr$^{-1}$ in the Western Alps and the Pyrenees, associated with radial shortening in the Western Alps foreland, compatible with previous studies (e.g., Rigo et al., 2015; Walpersdorf et al., 2018).
- In addition, several areas (Aquitaine Basin, Brittany, Bresse) present unexpected high deformation rates, some of them anti-correlated with the local level of seismicity.
- The only significant vertical velocity patterns are the subsidence in the Paris Basin (*ca.* 0.5–1 mm yr$^{-1}$) and uplift in the Western and Central Alps (*ca.* 0.5–2 mm yr$^{-1}$).

These results open questions about the origin and the time scales of these deformations, as well as their relationship with seismicity and the potential for significant aseismic deformation in France and Western Europe. In particular, these small velocity and deformations signals may relate to a variety of non-tectonic forces, such as Glacial Isostatic Adjustment or

hydrological loading, operating from a local (100–200 km) to a continental scale.

In addition to these observations, our methodological developments highlight two conclusions regarding the potential of future more detailed analyses:

- The station density is critical to eliminate outliers and extract spatially coherent deformation signals. Non-geodetic networks and campaign data, although of potentially lower resolution compared to permanent geodetic stations, can
provide important densification.
- A consistent processing of the raw (RINEX) data and position time series is essential to extract significant deformation. Our results for the Western Alps are currently limited by the post-processing combination of velocity solutions that result in uncertainties of *ca.* 50 km in the peak of the extension pattern.

Finally, the combination of mathematical technics for extracting spatially coherent signals can provide confidence, or point

out limitations, in the observed deformation patterns beyond the application of simple standard error estimations.

Data Availability





The data_providers file shows all the sources used to obtain the data present in this study. The time series analyses were performed using R (R Core Team, 2016). Maps were done with GMT5 (Wessel et al., 2011).

Author contributions.

5  CM and ED processed the GPS data. CM and SM developed the analysis methods. CM did the statistical analyses. CM, SM, and PV interpreted the results and wrote the article.

Competing interests.

The authors declare that they have no conflict of interest.

Acknowledgments

We are grateful to the technical teams of Montpellier and RENAG-RESIF. We deeply thank all data providers.

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

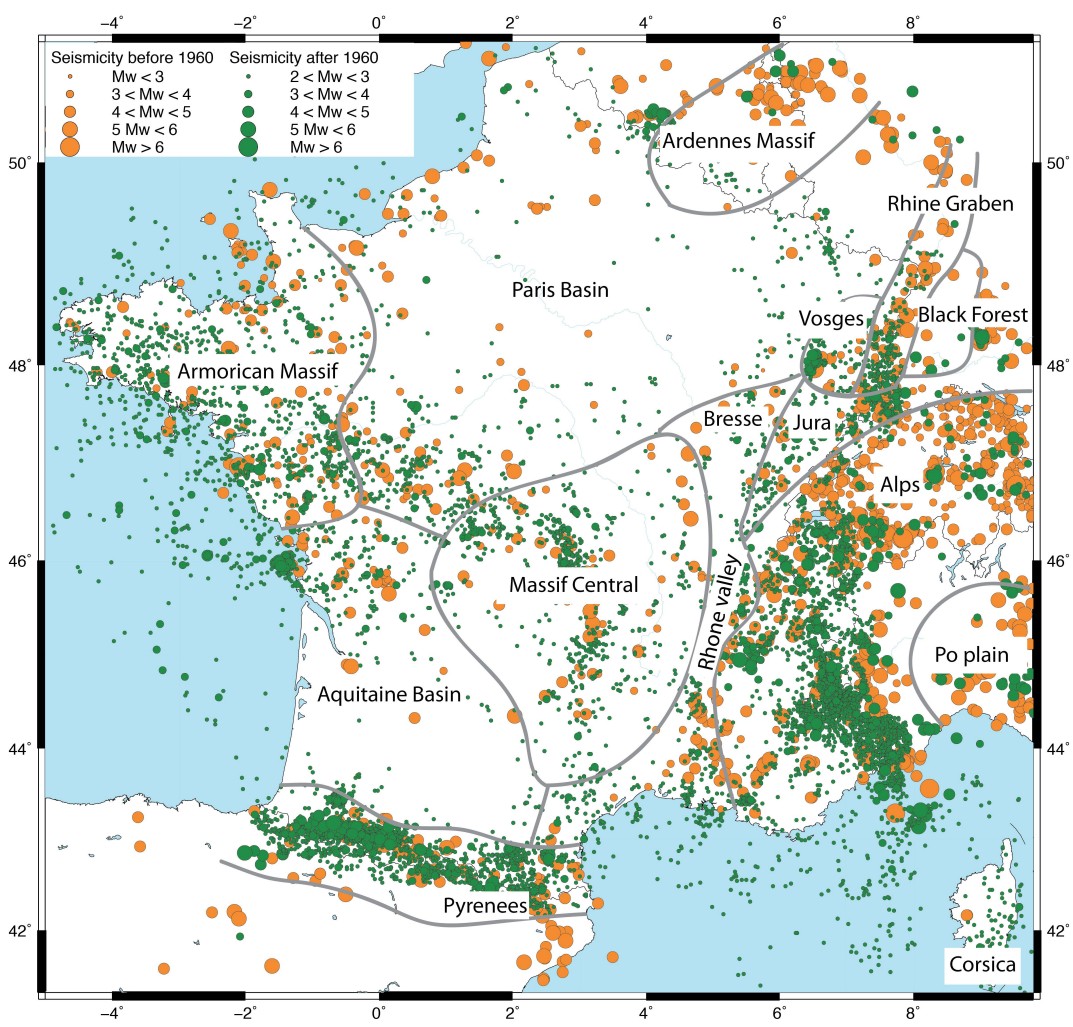

Figure 1: Map of instrumental in green and historical in orange seismicity (SHARE and SI-Hex). Grey lines delimit the main geographical or sometimes geological areas.

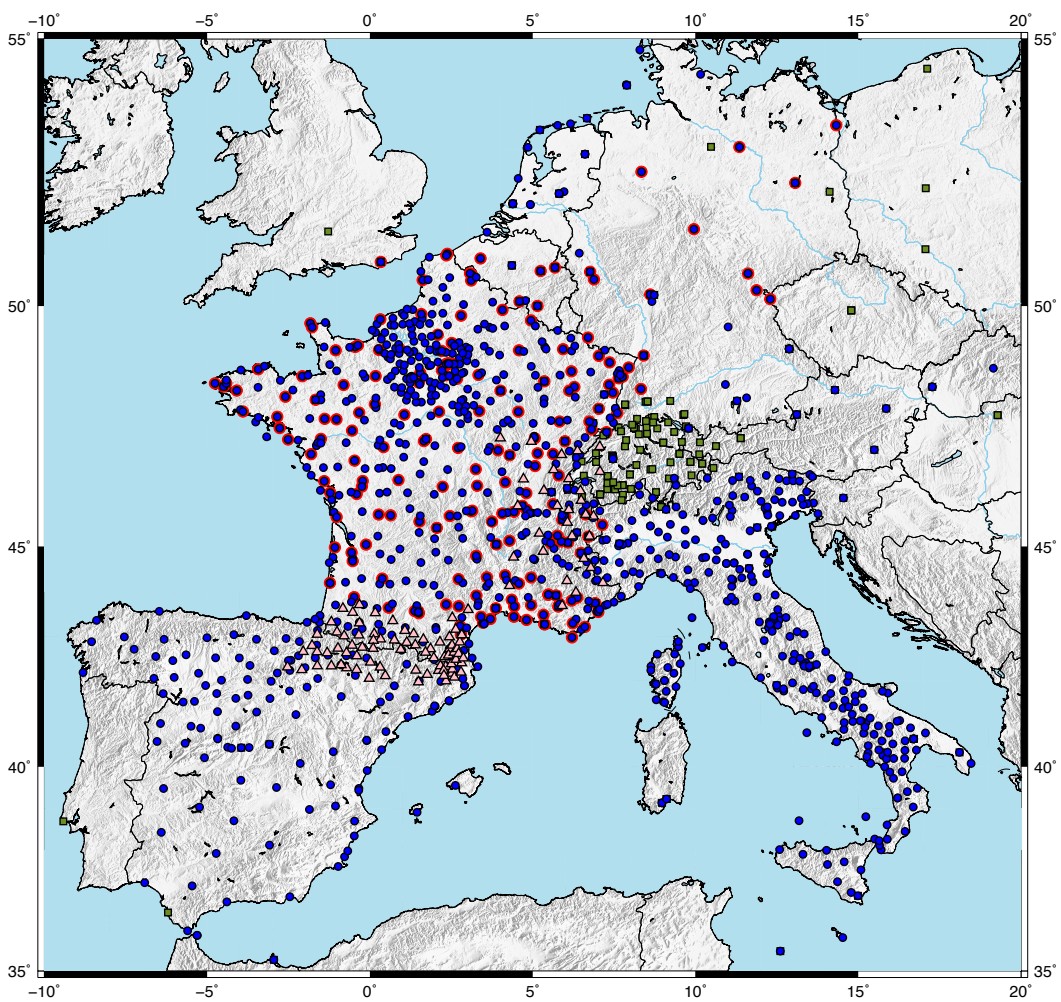

Figure 2: Distribution of the permanent (blue circles and blue circles with red outline for the stations used to define the "France-centered" reference frame) and campaign (pink triangles) stations used in this study. Green squares are combined Swiss stations.



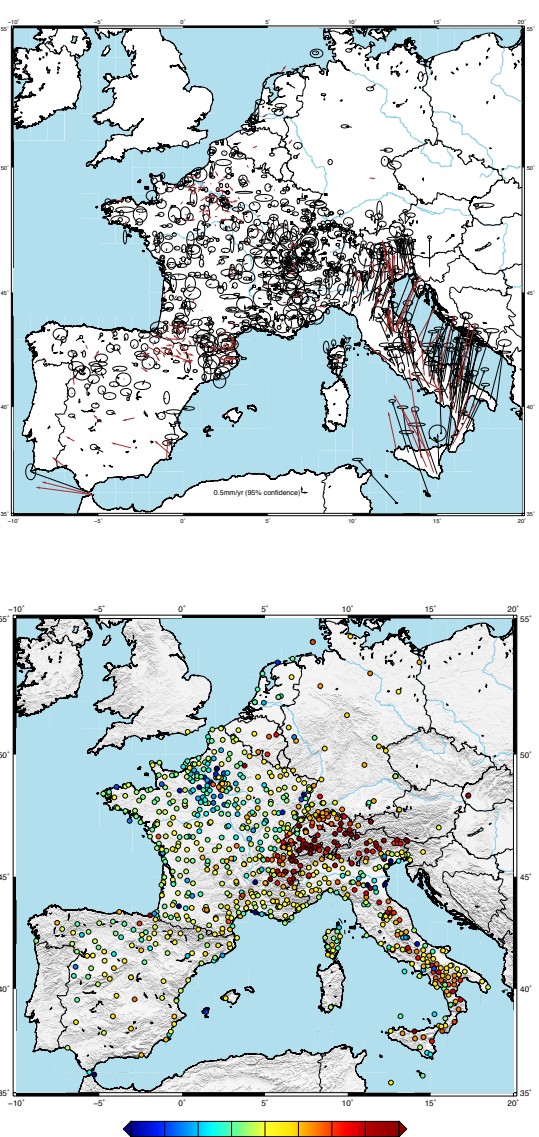

Figure 3: (a) Horizontal and (b) vertical velocities for permanent stations in France reference frame.





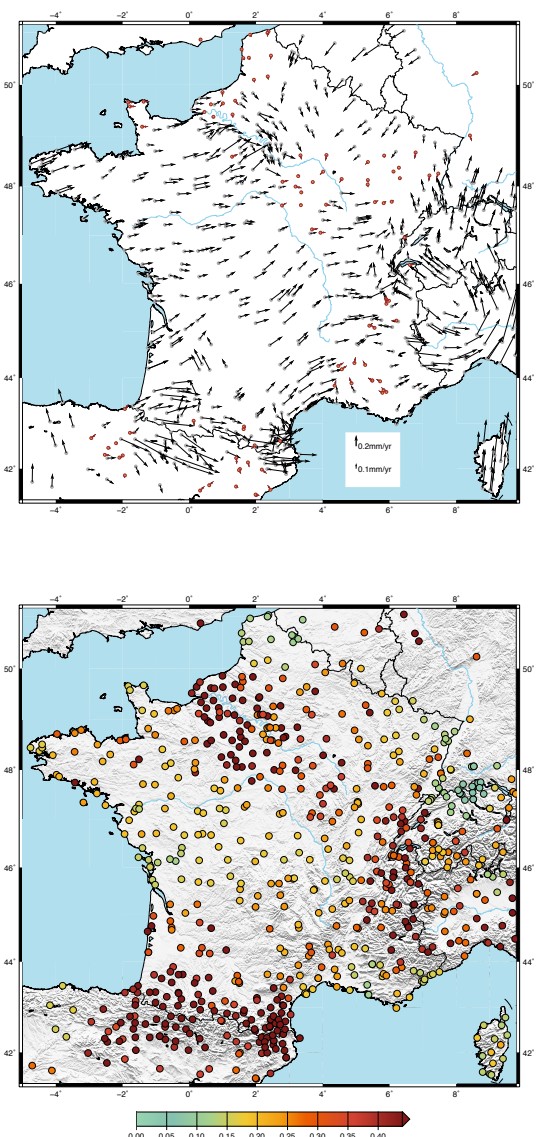

Figure 4: (a) Horizontal velocities from clustering with (b) 95% confidence interval (CI95).
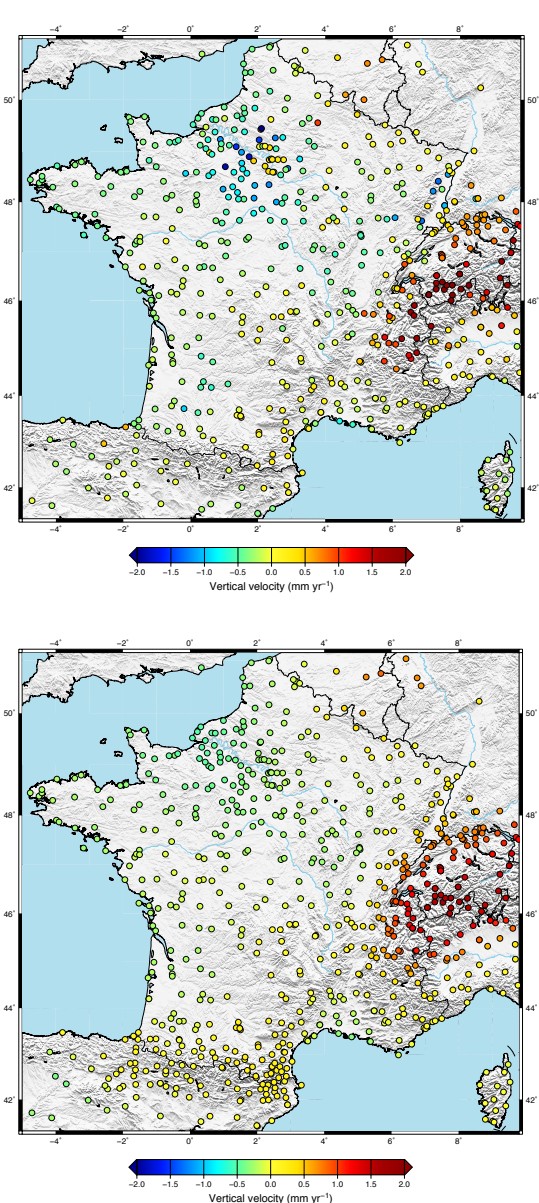

Figure 5: Vertical velocities (a) from clustering and (b) from Gaussian smoothing.


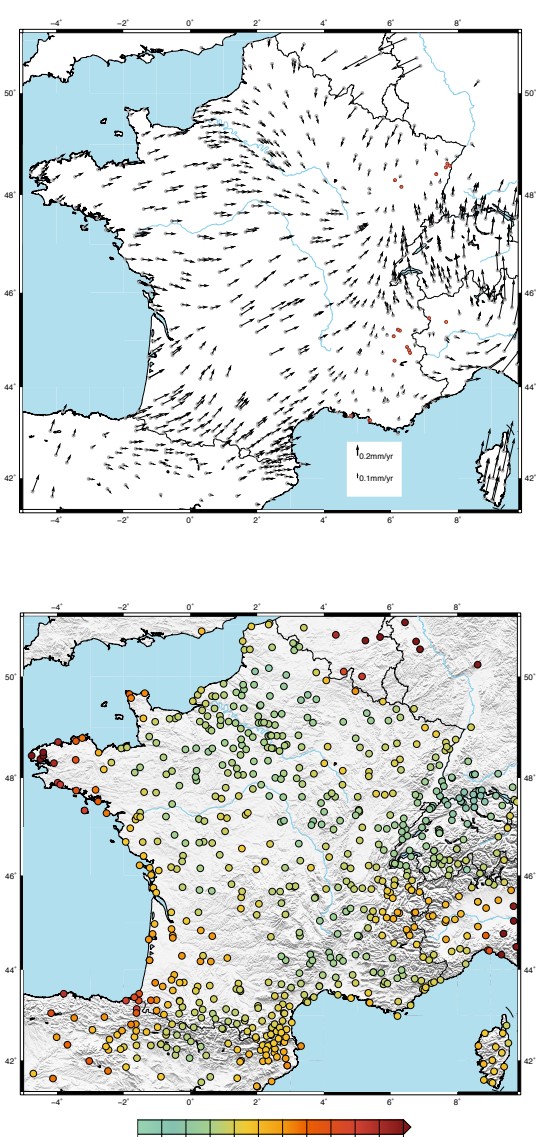

Figure 6: (a) Horizontal velocities for each site from Gaussian smoothing with (b) 95% confidence interval (CI95).

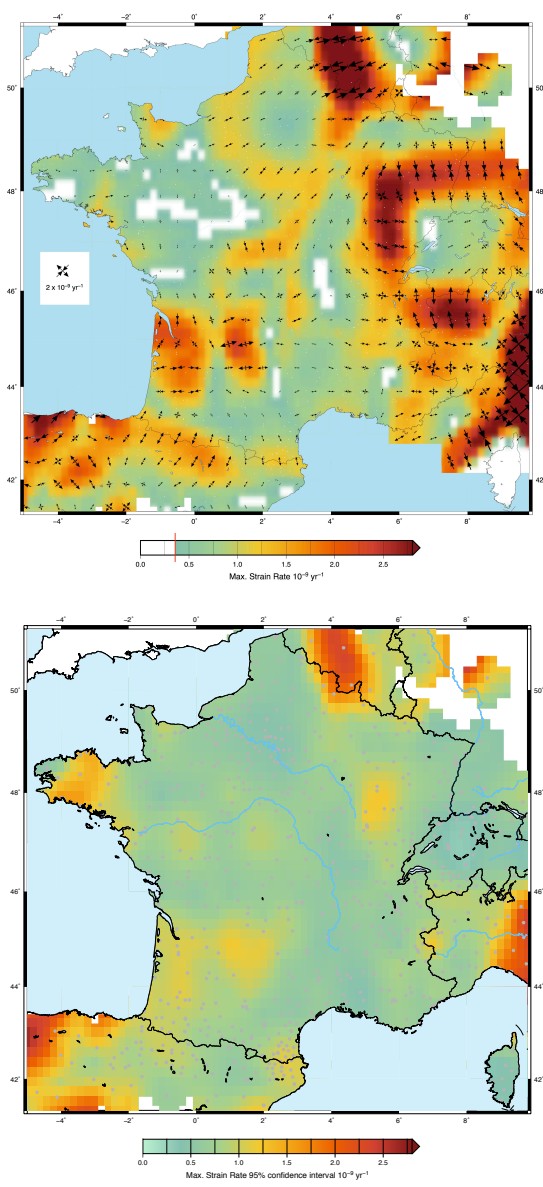

Figure 7: (a) Smoothed horizontal strain rate field with a grid of 0.1 degrees (strain rate tensors are plotted every 0.5 degrees) and (b) associated 95% confidence interval (CI95).




Figure 8: Map of the differences between the velocities obtained from the clustering and from the Gaussian smoothing methods.



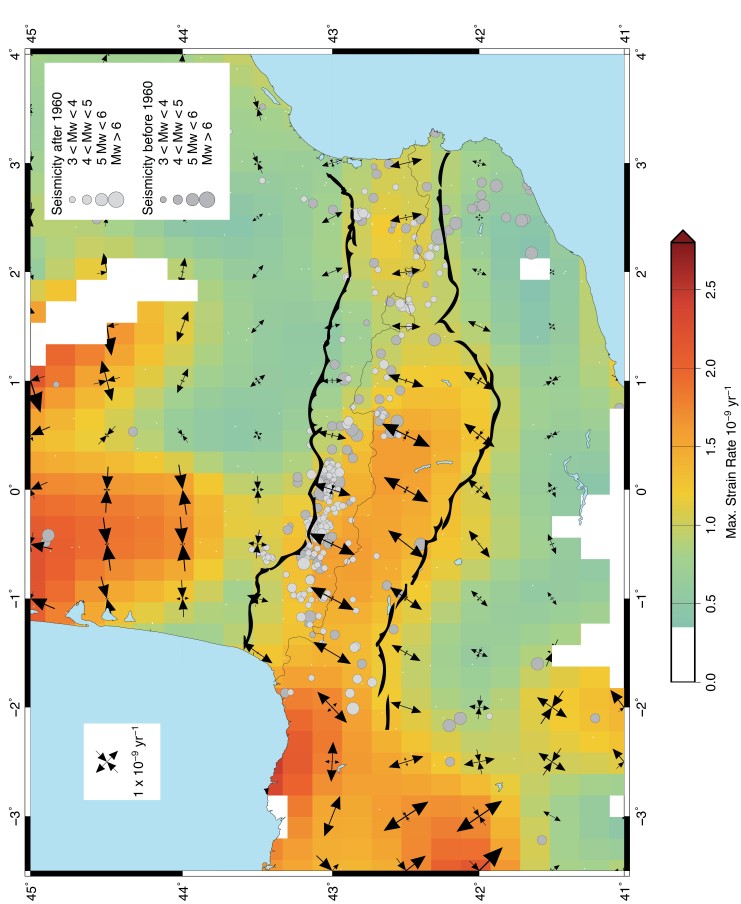

Figure 9: Smoothed horizontal strain rate field for Pyrenean permanent and campaign stations. With instrumental (dark gray) and historical (light gray) seismicity (SHARE and SI-Hex) and the main fault zones bordering the Pyrenees to the South and to the North.
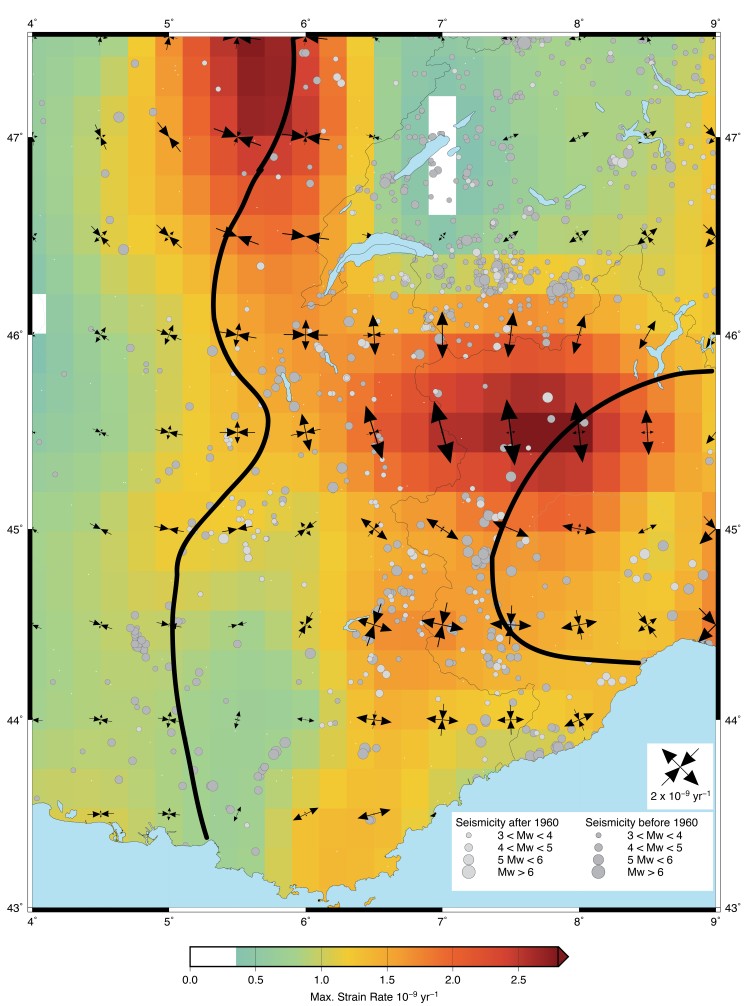

Figure 10: Smoothed horizontal strain rate field for Alpine permanent and campaign stations. With instrumental (dark gray) and historical (light gray) seismicity (SHARE and SI-Hex) and the limits between Alps and Po plain and Rhone valley.





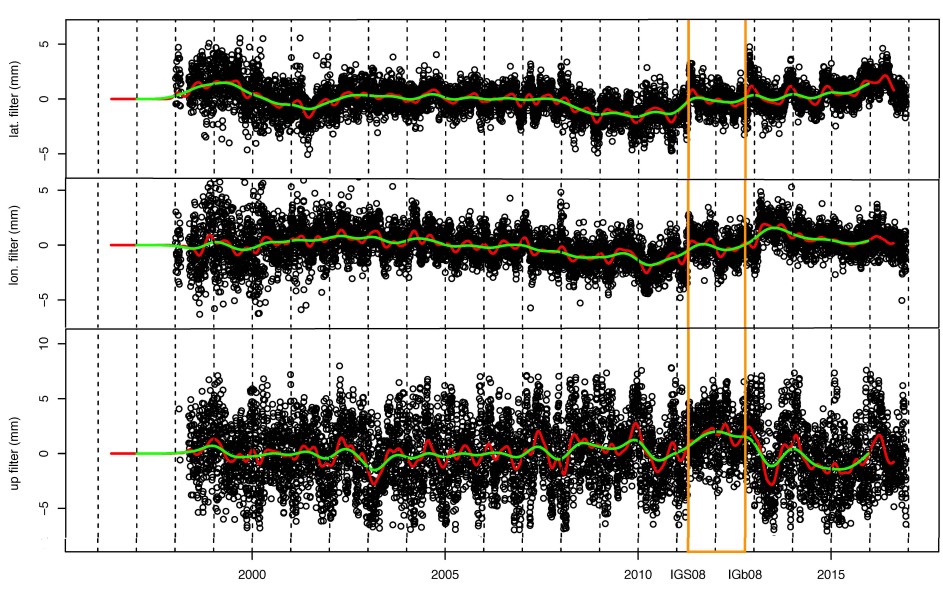

Figure A: Common mode stacked filter. Black points are the daily positions, red and green are the 1-month and 1-year time averages. Yellow lines are the IGS05 - IGS08 and IGS08 - IGb08 transitions.



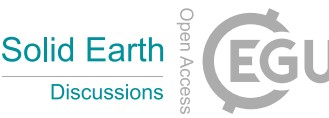

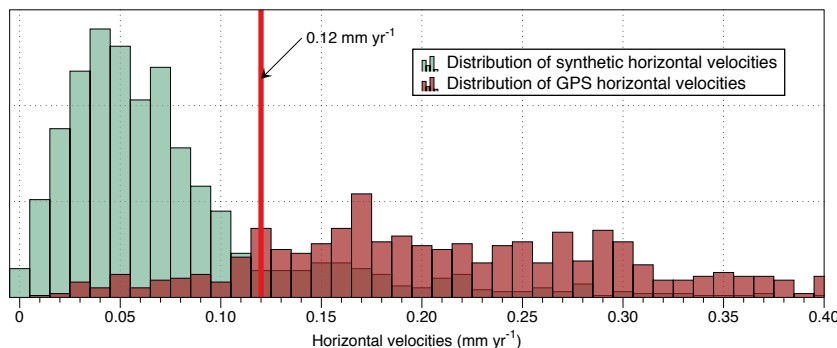

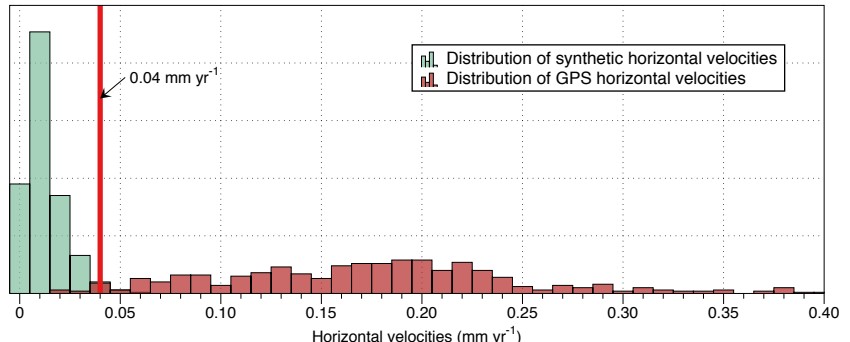

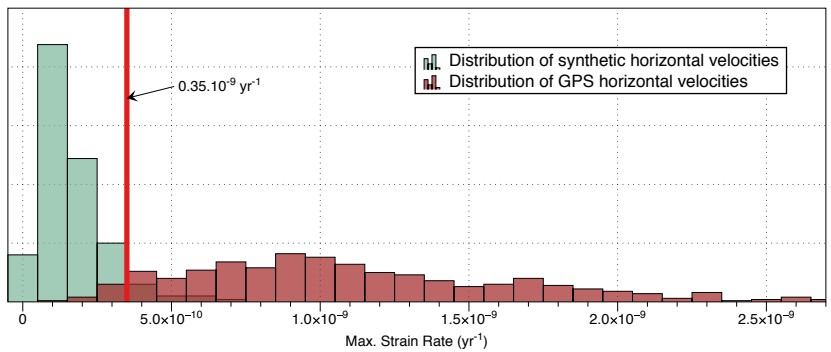

Figure B: Histograms of distribution of synthetic data vs. actual data for (a) clustered velocities, (b) smoothed velocities and (c) max strain rate.



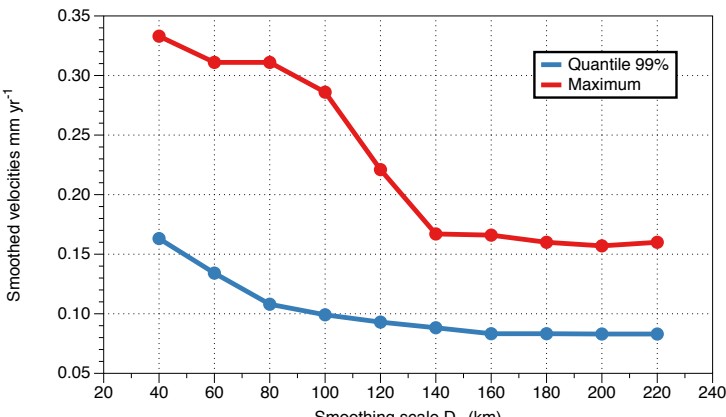

Figure C: Evolution of the smoothed velocities (maximum and quantile 99%) according to the smoothing distance.





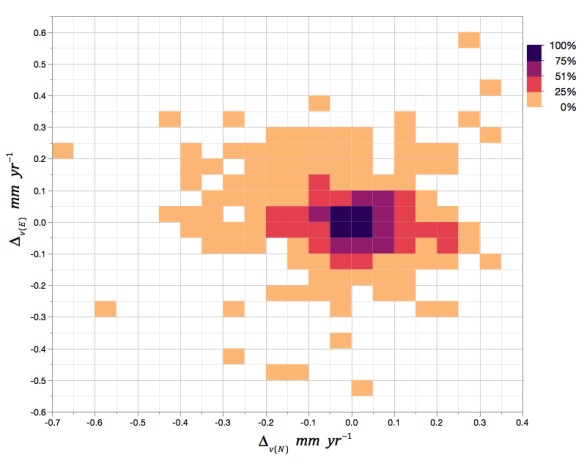

Figure D: Diagram of dispersion of di erential horizontal velocities between clustered and smoothed velocities, each color represents 25% of the totality of the data.