# Peer review of "Extracting small deformation beyond individual station precision from dense GNSS networks in France and Western Europe"

_Solid Earth, 2019_

## Referee Comment (RC1) · Anonymous Referee #1 · 23 Jun 2019

This paper presents a strain rate field model for the Greater France area estimated from GNSS-derived velocities. The inference of strain rates at or below the limit of the data precision is a challenging task and this is the reason why strain rate modeling for intraplate is difficult, at least at the spatial scales the authors aim at (∼100-200 km). The main issue is that outlier velocities can cause large strain rate artifacts, because strain rates are spatial derivatives, causing noise to amplify. I commend the authors for tackling this problem and for trying to create a "robust" solution. This study approaches the problem through a combination of cluster analysis (which replaces observed velocities with the dominant velocity for a local cluster (or so I understand)) and Gaussian smoothing. The resulting velocities are then used to infer spatial gradients and define

strain rates. The results look a bit puzzling with there being a number of significant elevated strain rate zones at places where they were unexpected (except for the Alps and Pyrenees). I am very concerned that the clustering approach, instead of having revealed systematic strain rate signals that were buried in the original data, has actually created signals that weren't there. The study also presents vertical rates but that analysis seems a bit disjointed from the rest of the paper.

I have many comments, listed below in descending order of approx. significance. Before I discuss those I want to point out that I was expected to see reference to the (Kreemer et al., 2018) paper. While the authors could in general do better with citing references (see comments below), that particular paper had the same aim as this study (to present a robust procedure to pull signal out of noisy data) but applied to intraplate North America. Perhaps the authors were unaware of that paper, and I strongly encourage them to check it out.

1) While I don't think I fully understand the clustering analysis, it seems to me that the resulting velocity field (Fig. 4a) is much more clustered than the original velocity field seen in Fig. 3a. My slight hesitation comes from the fact that a clear comparison is hard to make since the original data (Fig 2,3) is presented for a much larger geographic area (and a different scale) than the rest of the paper (see comment 8), which makes the observed velocities in the France area hard to see. Could the authors either reduce the area for Fig 2,3 or add a figure that shows observed velocities for the same geographic area as the other figures? In any case, Fig 4a is ultimately being used as input to the Gaussian smoothing, and it has various curious traits. Is it still in the same "France reference frame" as the data? If so, the fact that there is a dominant eastward component to most velocities suggests that the clustering changed the essential characteristic of the velocity field in this frame. How come? Obviously this velocity field is not in a new "France reference frame", because then that eastward motion should not be there. While the reference frame of the velocity field ultimately doesn't matter (because the purpose is to investigate strain rates, which are reference frame independent (although see comment 5), this seemingly change in reference frame by the cluster analysis points to a possible problem with the clustering analysis. Secondly, from Fig 4a it is clear that the clustering analysis broke up the velocity field in domains and that there are rather discrete boundaries between these domains. Some of the main features in the strain rate field (Fig 7a) are directly related to these cluster boundaries; the NS zone in Aquitaine Basin, the NW trending zone in the Paris Basin, the NS zone in northernmost France, and the three related zones in Eastern France: the EW compressional zone in NE France, the EW extensional zone in eastern France and the NS compressional zone that connects them.

2) I'll leave it up to the authors to find out what may be wrong with the clustering analysis, but I have two immediate suggestions that may further exemplify problems with the clustering: a) show the (vectorial) difference between the original velocities and those obtained from clustering. Right now the authors only show the difference between the velocities from the clustering and those from the subsequent smoothing (Fig. 8) and they don't show them vectorially, which is important. b) derive a strain rate model from the Gaussian smoothing but based instead on the original horizontal velocities. I expect many differences. While the authors may argue that those differences point to the clustering pulling out spatially coherent strain rate signals, I would argue that the clustering seemingly creates signals that are inconsistent with the original data.

3) Because of the concerns expressed above, I have little confidence in the validity of the observed strain rate features and the discussion thereof (Section 4 and 5). Part of the discussion is the comparison with seismicity. The authors indeed find no or confusing correlation (which the authors call "surprising"). The relationship between intraplate deformation deformation and seismicity is a hot science topic, and I am worried that that the general discussion on this topic does not benefit from comparisons being made on the basis of a strain rate model that has some serious problems. The authors also don't offer a good explanation for the various strain rate features in eastern France (except the Alps); I suggest this is because there isn't any good tectonic explanation and

that these features are modeling artifacts.

4) The authors don't question their results because they have faith in their uncertainties which they derived from a synthetic test in which strain rate model was inferred when observed velocities were nominally set to zero (but velocity uncertainties were kept). This may be a good test, as it shows how data uncertainty and network geometry map into model uncertainties. The clustering approach may also work well when velocities are set to zero, as it would be hard to make clusters out of such data. The clustering may however fail when it starts to determine median velocities from actual velocities.

5) I am quite confused by the strain rate estimation as part the Gaussian smoothing. Here are the reasons: a) it appears this is done on a flat-Earth approximation, which is may be ok, but given that this study tries to infer very small strain rates it is worth investigating what magnitude of error a flat-Earth approximation would introduce over a fully spherical treatment. b) equation (3) is quite similar to equation B2 of (Mazzotti et al., 2011 which they reference, but some curious differences exist: in the current study the azimuthal weighting function is missing (why?), and the velocity in the latter half of both components is here given as that of a station and in Mazzotti et al as that of the grid point (the latter appears correct). c) In general, I am puzzled how the strain rate field is parameterized as being the product of distance and velocity, because strain rate is ultimately related to velocity divided by distance. This explanation was missing in Mazzotti et al as well and I would suggest deriving and/or explaining this better. d) the velocities contain a translation/rotation (which is particularly a problem in light of the reference frame problem discussed in comment 1). The way it reads now is that any rotation gets mapped into strain rate. Please clarify. e) Is there any fundamental difference between this method and the VISR method of (Shen et al., 2015) or even the SSPX method of (Cardozo & Allmendinger, 2009)?

6) For the outlier detection, some questions came up: a) it is not mentioned, but are the detected outliers the red vectors in Fig 3a? b) How many of the added campaign velocities are identified as outliers? It seems like a lot. Is it still worth including those?

c) I don't understand line 22-23 (page 6) "Stations for which DM is greater than the network 95% confidence interval are considered as outliers and rejected". Does this mean that the outlier detection is based on distance as well? Why? d) Note that Kreemer et al. (2018) also introduced an algorithm to identify outliers. e) To test the "robustness" of the presented strain rate model one would need to show that the model is not affected by outlier data (if that is what was indeed meant with the model being "robust"). I understand why they flagged outliers, but to proof robustness they should also show a model that was based on data that included outlier velocities. Ideally the resulting model would be mostly the same.

7) the vertical velocities are also subjected to the clustering analysis and subsequent smoothing. The results are only sporadically mentioned in the discussion, which makes one wonder about that part of the presented data in light of this study's goals. There have been other recent attempts to obtain a "smooth" vertical velocity field (either as a continuous grid and/or by "despeckling" the original rates, as is done here). Examples are: (Hammond et al., 2016; Husson et al., 2018; Serpelloni et al., 2013) The authors should consider discussing and/or comparing the various approaches.

8) It is not clear why the authors present data over an area much larger than the ultimate study area. The study and most figures are focused on greater France but Figs. 2 and 3 show a much larger area, which notably includes a lot of data in Italy. Why is this presented if it isn't used? Does the number of stations mentioned in the text include those in Italy? If yes, I would find that misleading. While I understand that the authors would want to add a little buffer to the area show in most figures, the current presentation is confusing and doesn't allow for a good comparison between data and model in the actual study area.

9) In the introduction (line 21-22) of page 1, some studies of intraplate strain rate are mentioned (Canada, India). It would be better if the mentioned studies would be previous attempts to model intraplate strain rates in the same area, which are currently not even mentioned, particularly (Tesauro et al., 2006).

10) Abstract, first sentence. Authors say "we use dense geodetic networks and large GPS datasets". What is the distinction between these two? They seem the same.

11) The paper uses the word "technics" twice. Ironically, the English language uses the French word: "techniques". Please correct.

12) The details on the GPS data analysis do not mention the minimum duration of the considered time-series. Is it 2.5 years? If not, what it it? If less, why?

13) page 3, line 11: "only a small percentage of stations is associated with reliable equipment logs". I suppose this hinges on the word "reliable" but I would have thought that the majority of the stations would have logs.

14) page 3, line 15-16. Here the bias is mentioned of undetected jumps on velocities (I think) but only for long time-series (>8 years). How about short(er) time-spans?

15) Was the common-mode also removed from the campaign data. It should be, but wasn't explicitly mentioned

16) page 7. the authors say that a spatial scale of 100-200 km corresponds to the interseismic deformation on a (vertical?) fault with a seismogenic thickness of 10-25. Of course, that would totally depend on the slip rate (and the precision in the data), so I think it would be better to omit this statement.

17) what are the orange colored points in Fig 4a and Fig 6a?

18) With the chance of sounding like a curmudgeon; the last author's contribution is solely in the realm of GPS data processing. Does that warrant authorship? (Note that this comment is not affecting my assessment of this paper)

Response to formal review criteria:

Scientific significance: Fair Scientific Quality: Poor Presentation Quality: Fair

+ Does the paper address relevant scientific questions within the scope of SE? yes

Does the paper present novel concepts, ideas, tools, or data? yes

• Are substantial conclusions reached? No

• Are the scientific methods and assumptions valid and clearly outlined? No

• Are the results sufficient to support the interpretations and conclusions? No

• Is the description of experiments and calculations sufficiently complete and precise to allow their reproduction by fellow scientists (traceability of results)? No

• Do the authors give proper credit to related work and clearly indicate their own new/original contribution? No

• Does the title clearly reflect the contents of the paper? Yes

• Does the abstract provide a concise and complete summary? Yes

• Is the overall presentation well structured and clear? Yes/No

• Is the language fluent and precise? Yes/No

• Are mathematical formulae, symbols, abbreviations, and units correctly defined and used? Yes/No

• Should any parts of the paper (text, formulae, figures, tables) be clarified, reduced, combined, or eliminated? No

• Are the number and quality of references appropriate? Yes/No

• Is the amount and quality of supplementary material appropriate? Yes

Cited References:

Cardozo, N., & Allmendinger, R. W. (2009). SSPX: A Program to Compute Strain from Displacement/Velocity Data. Comput. Geosci., 35(6), 1343–1357. https://doi.org/10.1016/j.cageo.2008.05.008

[Figure]

Hammond, W. C., Blewitt, G., & Kreemer, C. (2016). GPS Imaging of vertical land motion in California and Nevada: Implications for Sierra Nevada uplift. Journal of Geophysical Research: Solid Earth, 121(10), 7681–7703. https://doi.org/10.1002/2016JB013458

Husson, L., Bodin, T., Spada, G., Choblet, G., & Kreemer, C. (2018). Bayesian surface reconstruction of geodetic uplift rates: Mapping the global fingerprint of Glacial Isostatic Adjustment. Journal of Geodynamics, 122, 25–40. https://doi.org/10.1016/j.jog.2018.10.002

Kreemer, C., Hammond, W. C., & Blewitt, G. (2018). A robust estimation of the 3-D intraplate deformation of the North American plate from GPS. Journal of Geophysical Research-Solid Earth, 123, 4388–4412. https://doi.org/10.1029/2017JB015257

Mazzotti, S., Leonard, L. J., Cassidy, J. F., Rogers, G. C., & Halchuk, S. (2011). Seismic hazard in western Canada from GPS strain rates versus earthquake catalog. Journal of Geophysical Research, 116(B12), B12310. https://doi.org/10.1029/2011JB008213

Serpelloni, E., Faccenna, C., Spada, G., Dong, D., & Williams, S. D. P. (2013). Vertical GPS ground motion rates in the Euro-Mediterranean region: New evidence of velocity gradients at different spatial scales along the Nubia-Eurasia plate boundary. Journal of Geophysical Research: Solid Earth, 118(11), 6003–6024. https://doi.org/10.1002/2013JB010102

Shen, Z.-K., Wang, M., Zeng, Y., & Wang, F. (2015). Optimal Interpolation of Spatially Discretized Geodetic DataOptimal Interpolation of Spatially Discretized Geodetic Data. Bulletin of the Seismological Society of America, 105(4), 2117–2127. https://doi.org/10.1785/0120140247

Tesauro, M., Hollenstein, C., Egli, R., Geiger, A., & Kahle, H.-G. (2006). Analysis of central western Europe deformation using GPS and seismic data. Journal of Geodynamics, 42(4–5), 194–209. https://doi.org/10.1016/j.jog.2006.08.001

---

## Author Comment (AC1) · 24 Jun 2019

The first review points out a potential misunderstanding in our methodology : We apply each method (clustering and smoothing) on the raw GPS velocities. The two methods are fully independent. The strain rate map is computed using the smoothing method only (based on the raw data), independently of the clustering analysis.

---

## Referee Comment (RC2) · Mimmo Palano (Referee) · 20 Jul 2019

The manuscript is well written and well organized.

Mathematical formulations are corrected and clearly quoted.

Quoted papers are all necessary.

My minor questions are so summarized:

The GNSS is a primary technique, not a dataset. Please correct.

The use of "raw GNSS data" is misleading, when speaking on velocity field. Usually,

such a term is referred to RINEX GNSS data. Please check the term.

I suggest to expand the first paragraph of introduction by explaining that crustal deformation at various temporal and areal scales are measured on volcanic areas also (see for instance Kilauea and Etna). In doing this you can also improve the discussion on the applicability of your approach on volcanic areas.

A table reporting velocity field both in ITRF2014 and the local reference frame should be added as supplementary material.

Figures are of good quality; however, they need some small corrections. Please add a "north symbol" and a km scale to the figures. Moreover, all the figures reporting the local seismicity contain an error on the legend.

---

## Author Comment (AC2) · 5 Sep 2019

RC1: 0) This study approaches the problem through a combination of cluster analysis (which replaces observed velocities with the dominant velocity for a local cluster (or so I understand)) and Gaussian smoothing. The resulting velocities are then used to infer spatial gradients and define strain rates. The results look a bit puzzling with there being a number of significant elevated strain rate zones at places where they were unexpected (except for the Alps and Pyrenees). I am very concerned that the clustering approach, instead of having revealed systematic strain rate signals that were buried in the original data, has actually created signals that weren't there. The study

also presents vertical rates but that analysis seems a bit disjointed from the rest of the paper. AC: Several of the following comments are related to a misunderstanding regarding the usage of clustered velocities in the strain rate computations. We may not have been clear enough on this point in the original manuscript, and we understand how this misunderstanding might have led the reviewer to question some of our results. In practice, the application of clustering has not biased the analysis of deformation rates because they are totally independent. Clustering only provides a velocity field and no stain rates are derived from clustering. The fact that smoothing and clustering velocity fields reveal the same patterns is therefore a very good thing. This is the main idea of this study: from 2 completely independent methods we can extract the same information regarding regional kinematics and deformation contained in a low and noisy velocities field. As a first answer, we have added an author comment during the review phase to clarify this point: "We apply each method (clustering and smoothing) on the raw GPS velocities. The two methods are fully independent. The strain rate map is computed using the smoothing method only (based on the raw data), independently of the clustering analysis." To clarify the manuscript and eliminate the risk of misunderstanding, we added this sentence: P7L21 "We apply each method (clustering and smoothing) on the GPS velocities. The two methods are fully independent. Clustering only provides a velocity field and no stain rate field. The strain rate map (Fig. 7) is computed using the smoothing method only (based on the GPS velocities), independently of the clustering analysis." Also we specified the titles of the sections. P8L7 "Clustering applied to GPS velocities" and P9L7 "Gaussian smoothing applied to GPS velocities".

RC1: I have many comments, listed below in descending order of approx. significance. Be- fore I discuss those I want to point out that I was expected to see reference to the (Kreemer et al., 2018) paper. While the authors could in general do better with citing references (see comments below), that particular paper had the same aim as this study (to present a robust procedure to pull signal out of noisy data) but applied to intraplate North America. Perhaps the authors were unaware of that paper, and I

strongly encourage them to check it out.

AC: Indeed, Kreemer et al. 2018 is similar to ours but we forgot to quote it, thank you for this remark. The methodology and results are similar even if the study areas and the geodynamic mechanisms involved are different. We added this reference in several places: P1L27 "The lengthening of time series and the increase in the number of stations makes it possible to better constrain deformation in the intraplate domains (e.g., Kreemer et al., 2018; Tarayoun et al., 2018)." P7L5 "Other methods of detection of outliers exist (e.g., Kreemer et al., 2018)." P15L15 "in North America, Kreemer et al. (2018) have shown that, to first order, geodetic deformation is not directly correlated to seismicity and that the link between long-term tectonic processes, transient processes (GIA), seismicity and geodetic deformation is not simple at the scale of the regional deformation (> 100 km)."

RC1: 1) While I don't think I fully understand the clustering analysis, it seems to me that the resulting velocity field (Fig. 4a) is much more clustered than the original velocity field seen in Fig. 3a. My slight hesitation comes from the fact that a clear comparison is hard to make since the original data (Fig 2,3) is presented for a much larger geographic area (and a different scale) than the rest of the paper (see comment 8), which makes the observed velocities in the France area hard to see. Could the authors either reduce the area for Fig 2,3 or add a figure that shows observed velocities for the same geographic area as the other figures?

AC: You are absolutely right, thank you. For clarity, we modified Figure 2 to show the extent of the network used, Figure 3 has been cropped and Figure B show the entire velocity field. In Figure 3, there are now 3 elements: - in black, the horizontal velocities associated with their 95% uncertainties, - in brown the horizontal velocities whose uncertainty is larger than 0.3 mm/yr (for which the uncertainties are not represented for graphic reasons), - in red, the horizontal velocities of the stations identified as outliers.

RC1: In any case, Fig 4a is ultimately being used as input to the Gaussian smoothing,

and it has various curious traits.

AC: This remark is linked to the misunderstanding addressed in point n ° 0 (the clustered velocities are not used in the Gaussian smoothing).

RC1: Is it still in the same "France reference frame" as the data? If so, the fact that there is a dominant eastward component to most velocities suggests that the clustering changed the essential characteristic of the velocity field in this frame. How come? Obviously this velocity field is not in a new "France reference frame", because then that eastward motion should not be there. While the reference frame of the velocity field ultimately doesn't matter (because the purpose is to investigate strain rates, which are reference frame independent (although see comment 5), this seemingly change in reference frame by the cluster analysis points to a possible problem with the clustering analysis. Secondly, from Fig 4a it is clear that the clustering analysis broke up the velocity field in domains and that there are rather discrete boundaries between these domains. Some of the main features in the strain rate field (Fig 7a) are directly related to these cluster boundaries; the NS zone in Aquitaine Basin, the NW trending zone in the Paris Basin, the NS zone in northernmost France, and the three related zones in Eastern France: the EW compressional zone in NE France, the EW extensional zone in eastern France and the NS compressional zone that connects them.

AC: This remark includes several points: a) The velocity fields seem to highlight a systematic movement towards the East that questions the definition of reference frame. b) This effect would be a clustering bias. c) This, and other clustering effects, would be at the origin of several deformation zones highlighted.

Reply to points b) and c): (as a reminder, the analyzes (clustering and smoothing) are totally independent, see point n ° 0). The two velocity fields (Fig. 4a and 6a) highlight the same eastward trend in western France, which eliminates the hypothesis of a potential bias due to the clustering method. Furthermore, these apparent eastward motions cannot result in regional (ca. 100-200 km scale) deformation patterns as

shown on the strain rate map, which is based on smoothing only and has no link with clustering. In addition, the clustering method produces, by definition, strong edges in the velocity field that could be due to the station spatial distribution. In our analysis of regional deformation, we are careful to not interpret these potential biases due to the network configuration. It is worth noting that the clustering method is not new and has been applied to GPS velocities in the last decades or so. We added a couple of references to present this (Savage and Simpson 2013; Ozdemir and Karslioglu, 2019).

Reply to point a): Indeed, both clustering and Gaussian velocity fields (Fig. 4a and 6a) show a systematic movement towards the E-NE. The Gaussian smoothing method, from which deformation rates are derived, is based on the assumption of a flat Earth. The presence of a residual rotation can theoretically bias the estimation of strain rates. However, the deformations from the velocity field before subtraction of the rotational movement are similar to those presented here. The differences between the 2 estimates yield a value of $0.12 \times 10^{-9}$ yr-1 for the 95th percentile. This means that even if the entire rotational movement remains present, the associated deformation is much lower (one order of magnitude) than our detection level. We can therefore consider that the hypothetical presence of the rotation does not lead to significant bias in the determination of deformation rates. Although it does not bias the estimation of deformation rates, this movement must be explained. Its origin is linked on one hand to the stations used to define the rotational movement (stations of more than 7 years) and on the other hand to the fact that the stations of less than 7 years undergo a strong correction coming from the stack. Indeed, in the stack (Fig. A), we observe that in the last 7 years the trend slope shows a significant changes, notably because of the 2 changes of IGS frame in 2011 and 2012. The use of REPRO2 in a future calculation may allow overcoming these offsets and reduce the correction brought to the series of less than 7 years. NB: Such analysis requires much more details that are beyond the scope of this manuscript, further information can be found the C. Masson Ph.D. thesis (manuscript to be deposited in the HAL archive in late 2019).

RC1: 2) I'll leave it up to the authors to find out what may be wrong with the clustering analysis, but I have two immediate suggestions that may further exemplify problems with the clustering: a) show the (vectorial) difference between the original velocities and those obtained from clustering.

RC1: Right now the authors only show the difference between the velocities from the clustering and those from the subsequent smoothing (Fig. 8) and they don't show them vectorially, which is important.

AC: These vector differences have been added in Appendix and are mentioned in the manuscript (P11, L11). These maps confirm what is shown on Figs. 8 (differences in scalar amplitude), which indicate that the clustering does not show significant regional biases.

RC1: b) derive a strain rate model from the Gaussian smoothing but based instead on the original horizontal velocities. I expect many differences. While the authors may argue that those differences point to the clustering pulling out spatially coherent strain rate signals, I would argue that the clustering seemingly creates signals that are inconsistent with the original data.

AC: These remarks are linked to the misunderstanding addressed in point n ° 0.

RC1: 3) Because of the concerns expressed above, I have little confidence in the validity of the observed strain rate features and the discussion thereof (Section 4 and 5). Part of the discussion is the comparison with seismicity. The authors indeed find no or confusing correlation (which the authors call "surprising"). The relationship between intraplate deformation and seismicity is a hot science topic, and I am worried that that the general discussion on this topic does not benefit from comparisons being made on the basis of a strain rate model that has some serious problems. The authors also don't offer a good explanation for the various strain rate features in eastern France (except the Alps); I suggest this is because there isn't any good tectonic explanation and that these features are modeling artifacts.

AC: This remark is linked to the misunderstanding addressed in point n ° 0. Considering that the misunderstanding has been resolved, the reviewer points out that few interpretations have been made for the East of France. Indeed, we can develop the interpretation of the results in the East of France. Unlike the reviewer, we think that these results can have tectonic explanations. To clarify this point we added these elements to the Discussion: P14L1 "These results are consistent with those reported in other geodetic studies (Sanchez et al., 2018)." P14L14 "For the regions of the Vosges and Jura, the analysis of focal mechanisms provides results compatible with our geodesic deformations (Plenefisch et al. al., 1997, Maurer et al., 1997, Sue et al., 1999 and 2007). The compatibility suggests a tectonic origin of this deformation."

We would also like to point out that deeper analyses of our geodetic results in each area would require a lot more space than available in a SE manuscript. The main point of our study is to highlight the new applications and methods, hopefully triggering more complete analyses in the future. Some of the deformation patterns identified here might, in time, prove out to be incorrect, but this is part of the research process and should not deter the publication of new analyses and results. P2L18 "These interpretations are preliminary and for several regions a specific study is necessary."

RC1: 4) The authors don't question their results because they have faith in their uncertainties which they derived from a synthetic test in which strain rate model was inferred when observed velocities were nominally set to zero (but velocity uncertainties were kept). This may be a good test, as it shows how data uncertainty and network geometry map into model uncertainties. The clustering approach may also work well when velocities are set to zero, as it would be hard to make clusters out of such data. The clustering may however fail when it starts to determine median velocities from actual velocities. AC: It is quite difficult to address this comment.

Concerning the last point, the clustering analysis does not depend on the velocity amplitudes because it is essentially driven by the station coordinates to derive geographical groups (as explained in the manuscript). Thus, the tests with zero velocities are

representative of the real velocity analysis.

We would like to point out that we don't "have faith in the uncertainties". We propose a detailed analysis of uncertainties and resolution (much more detailed than most geodetic deformation studies). This analysis yields standard errors and detection levels that we use to define which parts of our results are significant and which are not. Our interpretation of what is significant is not subjective but factual (providing our computations and results are correct).

RC1: 5) I am quite confused by the strain rate estimation as part the Gaussian smoothing. Here are the reasons: a) it appears this is done on a flat-Earth approximation, which is may be ok, but given that this study tries to infer very small strain rates it is worth investigating what magnitude of error a flat-Earth approximation would introduce over a fully spherical treatment. AC: The answer to this remark is in point 1a. RC1: b) equation (3) is quite similar to equation B2 of (Mazzotti et al., 2011 which they reference, but some curious differences exist: in the current study the azimuthal weighting function is missing (why?), and the velocity in the latter half of both components is here given as that of a station and in Mazzotti et al as that of the grid point (the latter appears correct). AC: The azimuthal weighting is not used here because this feature is not very robust and results in additional complexities in the analysis, with little effect in the end. Concerning the grid point, we tried to clarify the equation to point out it could apply to a grid or only given point. This was not clear and we corrected this to limit to a grid analysis (as used here). RC1: c) In general, I am puzzled how the strain rate field is parameterized as being the product of distance and velocity, because strain rate is ultimately related to velocity divided by distance. This explanation was missing in Mazzotti et al as well and I would suggest deriving and/or explaining this better. AC: The fact that the distance comes as a multiplication is because of the derivative of the Gaussian exponential. It is a straight derivative, which we do not think requires a detailed presentation. However, if necessary, we can add it in a Supplementary Material. We will comply with the editor's opinion on this point. RC1: d) the velocities contain a

translation/rotation (which is particularly a problem in light of the reference frame problem discussed in comment 1). The way it reads now is that any rotation gets mapped into strain rate. Please clarify. AC: The answer to this remark is in point 1a (the Flat Earth bias is not significant, one order of magnitude smaller than the signal). RC1: e) Is there any fundamental difference between this method and the VISR method of (Shen et al., 2015) or even the SSPX method of (Cardozo & Allmendinger, 2009)? AC: SSPX uses a similar method. Discussing the differences between our approach and the numerous other methods that exist (beyond SSPX and VISR) is beyond the scope of our manuscript.

RC1: 6) For the outlier detection, some questions came up: a) it is not mentioned, but are the detected outliers the red vectors in Fig 3a?

AC: The legend of the figure was indeed not clear and has been modified (see point n ° 1).

RC1: b) How many of the added campaign velocities are identified as outliers? It seems like a lot. Is it still worth including those?

AC: There was a misunderstanding because of the legend of the figure (see point n ° 1). The identification of outliers was made only on permanent stations and horizontal components. We added a clarification P6L23 : "The statistical outlier detections are applied only on the horizontal velocities of permanent stations because vertical velocities show too much variability for robust results. Campaign stations are not included because of their large associated uncertainties."

RC1: c) I don't understand line 22-23 (page 6) "Stations for which DM is greater than the network 95% confidence interval are considered as outliers and rejected". Does this mean that the outlier detection is based on distance as well? Why?

AC: There was a misunderstanding: it is not a spatial distance but a statistical call. We added a clarification P7L1: "The term distance (DM) does not refer to a spatial distance

but to a statistical distance between each station variables and the barycentre of the multidimensional space formed by the network variables."

RC1: d) Note that Kreemer et al. (2018) also introduced an algorithm to identify outliers.

AC: Indeed, and the comparison of the results will be very interesting. We added this suggestion P7L5: "Other methods of detection of outliers exist (e.g., Kreemer et al., 2018)."

RC1: e) To test the "robustness" of the presented strain rate model one would need to show that the model is not affected by outlier data (if that is what was indeed meant with the model being "robust"). I understand why they flagged outliers, but to proof robustness they should also show a model that was based on data that included outlier velocities. Ideally the resulting model would be mostly the same.

AC: We added this figure on Supplementary Materials. In text P10L14 "(The smoothed horizontal strain rate fields but without excluding outliers is presented in Figure E)."

RC1: 7) the vertical velocities are also subjected to the clustering analysis and subsequent smoothing. The results are only sporadically mentioned in the discussion, which makes one wonder about that part of the presented data in light of this study's goals. There have been other recent attempts to obtain a "smooth" vertical velocity field (either as a continuous grid and/or by "despeckling" the original rates, as is done here). Examples are: (Hammond et al., 2016; Husson et al., 2018; Serpelloni et al., 2013) The authors should consider discussing and/or comparing the various approaches. AC: We discuss our vertical velocities, to a first order, but choose to focus on horizontal signal that we deem more interesting / novel. A more detailed analysis of the vertical results and comparisons with others studies would be interesting, but would likely require a specific study and manuscript.

RC1: 8) It is not clear why the authors present data over an area much larger than

the ultimate study area. The study and most figures are focused on greater France but Figs. 2 and 3 show a much larger area, which notably includes a lot of data in Italy. Why is this presented if it isn't used? Does the number of stations mentioned in the text include those in Italy? If yes, I would find that misleading. While I understand that the authors would want to add a little buffer to the area show in most figures, the current presentation is confusing and doesn't allow for a good comparison between data and model in the actual study area. AC: This remark was considered and addressed in points 1 and 6. And an explanation of the number of stations used has been added. P2L20: "The calculation was extended to a wider area than the frame considered in the rest of this study. Thus 313 stations mainly in Italy and Spain do not appear on France centered maps. Table S1 shows the velocities of all stations used in the calculation."

RC1: 9) In the introduction (line 21-22) of page 1, some studies of intraplate strain rate are mentioned (Canada, India). It would be better if the mentioned studies would be previous attempts to model intraplate strain rates in the same area, which are currently not even mentioned, particularly (Tesauro et al., 2006). AC: Indeed, we did not mention Tesauro et al. (2006) because we preferred to cite more recent studies. The study of Tesauro et al. (2006) is carried out on only 8 years of GPS data and with a rather heterogeneous network of stations. Despite the quality of this study and the results compatible with ours, we have privileged recent geodetic studies.

We now added this reference: P2L11 "Indeed, they are often considered as a domain without significant deformation except in its bordering mountain ranges, the Alps (e.g., Houlié et al., 2018; Brockmann et al., 2012; Tesauro et al., 2006) and the Pyrenees (e.g., Neres et al., 2018; Rigo et al., 2015)."

RC1: 10) Abstract, first sentence. Authors say "we use dense geodetic networks and large GPS datasets". What is the distinction between these two? They seem the same. AC: Exactly, we replaced these sentences (P1L6) with "We use two decades of data from a dense geodetic network to extract regionally coherent velocities and deformation rates in France and neighboring Western Europe." RC1: 11) The paper uses the word

"technics" twice. Ironically, the English language uses the French word: "techniques". Please correct. AC: Thank you for identifying these errors, we corrected them. RC1: 12) The details on the GPS data analysis do not mention the minimum duration of the considered time-series. Is it 2.5 years? If not, what it it? If less, why? AC: We have clarified this point. (P2L23) "The time series cover time spans from 1.5 to 19 years with an average duration of 7 years." RC1: 13) page 3, line 11: "only a small percentage of stations is associated with reliable equipment logs". I suppose this hinges on the word "reliable" but I would have thought that the majority of the stations would have logs. AC: To clarify this point, we have reformulated this sentence. (P3L13) "Because many stations are associated with incomplete equipment logs that could provide position off-set dates, the dates of potential offsets are automatically detected according to the method described in Masson et al. (2019)." RC1: 14) page 3, line 15-16. Here the bias is mentioned of undetected jumps on velocities (I think) but only for long time-series (>8 years). How about short(er) time-spans? AC: To answer this question, we developed the sentences about the results obtained by Masson et al., 2019. (P3L18) "Overall, us-ing this method, results in horizontal (resp. vertical) velocity biases are smaller than 0.2 mm yr-1 (resp. 0.5 mm yr-1) at 95% confidence levels for series longer than 8 years. For series with duration between 4.5 and 8 years and no offset, the velocity biases are smaller than 0.3 mm yr-1 (horizontal and vertical) at 95% confidence levels and, if at least one offset is present, the velocity biases are 0.6 mm yr-1 (resp. 1.3 mm yr-1). For the shortest series (less than 4.5 years), the velocity biases are larger than 1.0 mm yr-1 (Masson et al., 2019)." RC1: 15) Was the common-mode also removed from the campaign data. It should be, but wasn't explicitly mentioned AC: This point has been clarified. (P5L24) "Since the data are sporadic (a few points every 4–10 years), it is impossible to model annual and semi-annual seasonal signals, detect offsets and esti-mate noise characteristics by spectral analysis. No common-mode has been removed from the campaign data as the effect is not significant (Tarayoun et al., 2018)." RC1: 16) page 7. the authors say that a spatial scale of 100-200 km corresponds to the interseismic deformation on a (vertical?) fault with a seismogenic thickness of 10-25.

Of course, that would totally depend on the slip rate (and the precision in the data), so I think it would be better to omit this statement. AC: No, the spatial scale of the interseismic loading does not depend on the slip rate, only on the locking depth (cf. eq. 1 in Savage and Bufford, 1973). The detectability of such signal does depend on the slip rate, but the point here is that the spatial scale. But to simplify, this statement has been removed, thank you. (P7L21) RC1: 17) what are the orange colored points in Fig 4a and Fig 6a? AC: Sorry for the misunderstanding. The brown vectors correspond to those whose amplitude is lower than the detection levels determined by the synthetic data. The captions of Figures 4a and 6a have been clarified. RC1: 18) With the chance of sounding like a curmudgeon; the last author's contribution is solely in the realm of GPS data processing. Does that warrant authorship? (Note that this comment is not affecting my assessment of this paper) AC: We understand the remark, however Erik Doerflinger participated several months in the smooth running of the calculation. We would like to leave him in the list of co-authors to also value the work of technical staff. We will comply with the editor's opinion on this point.

---

## Author Comment (AC3) · 5 Sep 2019

RC2: The GNSS is a primary technique, not a dataset. Please correct.
The use of "raw GNSS data" is misleading, when speaking on velocity field. Usually, such a term is referred to RINEX GNSS data. Please check the term.

AC: Thank you for identifying this inconsistency. The use of terms has been standardized. P5L5, P14L22, P14L25: "the raw GNSS data (RINEX)" and P2L5 the incorrect terms have been replaced by "The GPS velocities".

RC2: I suggest to expand the first paragraph of introduction by explaining that crustal

deformation at various temporal and areal scales are measured on volcanic areas also (see for instance Kilauea and Etna). In doing this you can also improve the discussion on the applicability of your approach on volcanic areas.

AC: As suggested, we added mentions to volcanic studies in the introduction (P1L22) and the conclusion (P15L21) to expend the scope of our analyses.

RC2: A table reporting velocity field both in ITRF2014 and the local reference frame should be added as supplementary material.

AC: Thank you, indeed this essential table has been forgotten. It is now present in the Supplement Materials.

RC2: Figures are of good quality; however, they need some small corrections. Please add a "north symbol" and a km scale to the figures. Moreover, all the figures reporting the local seismicity contain an error on the legend.

AC: Indeed, it lacks a character, thank you. The figures have been modified. Orientation and scale were added in Figure 1.